# Generalized Probabilistic Approximate Optimization Algorithm

Abdelrahman S. Abdelrahman ●[1] ✉, Shuvro Chowdhury ●[1], Flaviano Morone ●[2] & Kerem Y. Camsari ●[1] ✉

We introduce the generalized Probabilistic Approximate Optimization Algorithm (PAOA), a classical variational Monte Carlo framework that extends and formalizes the recently introduced PAOA, enabling parameterized and fast sampling on present-day Ising machines and probabilistic computers. PAOA operates by iteratively modifying the couplings of a network of binary stochastic units, guided by cost evaluations from independent samples. We establish a direct correspondence between derivative-free updates and the gradient of the full Markov flow over the exponentially large state space, showing that PAOA admits a principled variational formulation. Simulated annealing emerges as a limiting case under constrained parameterizations, and we implement this regime on an FPGA-based probabilistic computer with on-chip annealing to solve large 3D spin-glass problems. Benchmarking PAOA against QAOA on the canonical 26-spin Sherrington–Kirkpatrick model with matched parameters reveals superior performance for PAOA. We show that PAOA naturally extends simulated annealing by optimizing multiple temperature profiles, leading to improved performance over SA on heavy-tailed problems such as SK–Lévy.

Monte Carlo algorithms remain a central tool for exploring complex energy landscapes, especially in the context of combinatorial optimization and statistical physics. Classical methods such as simulated annealing (SA) have been widely applied across these domains, but their reliance on slowly equilibrating processes limits their performance on rugged energy landscapes[1–6]. New approaches are needed to construct non-equilibrium strategies that retain algorithmic simplicity while improving solution quality.

Inspired by the quantum approximate optimization algorithm (QAOA)[7–17], Weitz et al.[18] proposed a classical variational protocol based on the direct parameterization of low-dimensional Markov transition matrices. Their work introduced the term Probabilistic Approximate Optimization Algorithm (PAOA), raising the possibility of classical, variational analogs to QAOA within probabilistic architectures.

Building on this foundational work, we formalize and generalize PAOA. We move beyond the original proposal's edge-local matrices to derive a global $2^N \times 2^N$ Markov-flow formulation applicable to any $k$-local Ising Hamiltonian of size $N$. This framework unifies a wide spectrum of variational ansätze, from global and local schedules to fully-parameterized couplings, and allows them to be stacked to an arbitrary depth $p$. Crucially, we connect this variational theory to practice by building on the p-computing framework[19], where networks of binary stochastic units (p-bits) sample from Boltzmann-like distributions through asynchronous dynamics. This model has a demonstrated record of success in optimization and inference tasks[20–23] and provides a natural substrate for our work. By implementing these dynamics on an FPGA, we demonstrate for the first time a scalable, hardware-based path for executing the PAOA.

We show that PAOA admits a broad class of parameterizations, including global, local, and edge-specific annealing schedules. Within this framework, SA emerges as a limiting case under constrained schedules. We implement this regime on an FPGA-based p-computer to demonstrate large-scale, high-throughput sampling. In contrast to standard SA, PAOA's flexible parameterization enables the discovery of non-equilibrium heuristics that exploit structural features of the problem. One such example is demonstrated in heavy-tailed

[1]Department of Electrical and Computer Engineering, University of California, Santa Barbara, Santa Barbara, CA, USA. [2]Department of Physics, Center for Quantum Phenomena, New York University, New York, NY, USA. ✉e-mail: abdelrahman@ucsb.edu; camsari@ucsb.edu

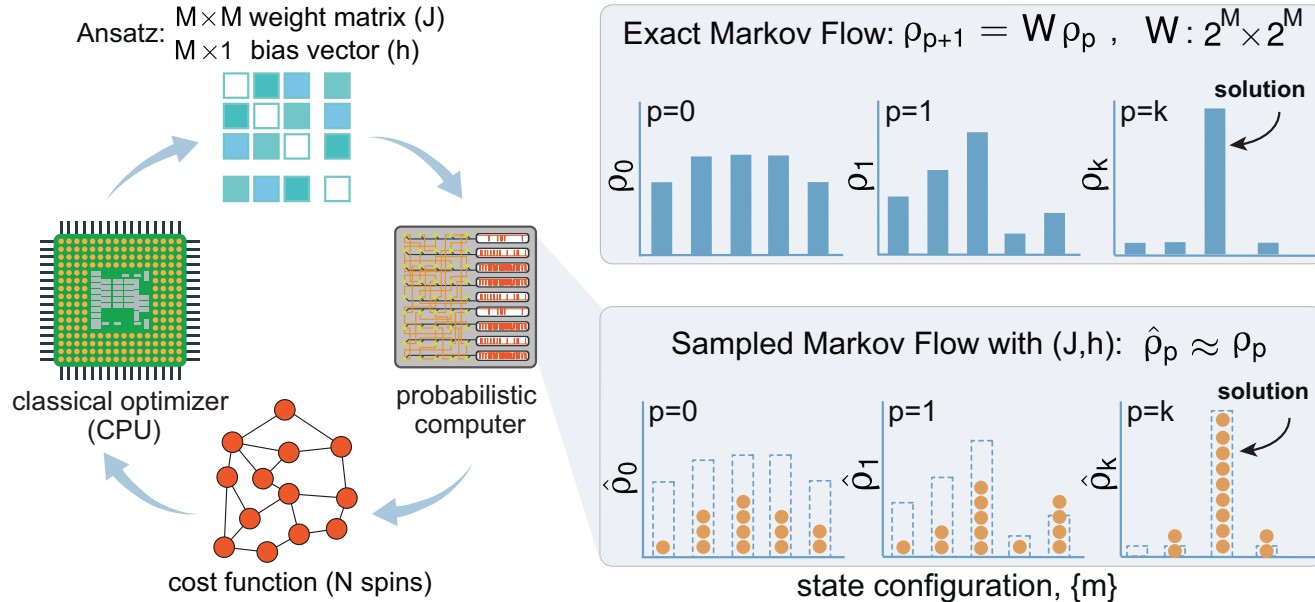

**Fig. 1 | Overview of PAOA.** A hybrid classical–probabilistic architecture iteratively updates a weight matrix $J$ and bias vector $h$ using feedback from a probabilistic computer. The p-computer samples from a distribution defined by $(J, h)$, approximating the exact Markov flow. The resulting samples $\hat{\rho}_p$ are used to evaluate a cost function, which the classical optimizer minimizes by adjusting the variational parameters. Here, $\rho$ denotes the exact probability distribution over spin configurations, $\hat{\rho}$ the sampled approximation, $p$ the layer number, $M$ the number of total spins represented in the ansatz (including possible hidden variables), and $\{m\}$ a specific spin configuration (state). The cost function is typically the energy of a spin glass mapped to an optimization problem (e.g., Eq. (3)) but can also be a likelihood function if PAOA is learning from data (e.g., Eq. (5)). Importantly, the ansatz size $M$ can exceed the original problem size $N$ since hidden variables may be introduced to increase the representation. In this work, we use $M = N$ in all experiments unless noted. At convergence ($p = k$), the distribution concentrates around low-energy solutions.

Sherrington–Kirkpatrick (SK) models, where PAOA assigns higher effective temperatures to strongly coupled nodes and lower temperatures to weakly connected ones, enabling annealing profiles with better performance over vanilla SA. This adaptive scheduling capability provides a powerful framework for the automated discovery of novel annealing heuristics.

We also benchmark PAOA against QAOA on the SK model using matched parameter counts and observe superior performance. This may not be surprising, as the practical performance advantages of QAOA over classical and probabilistic alternatives remain uncertain. A clear quantum advantage is expected to emerge in problems where interference plays an explicit role, such as in the synthetic examples constructed by Montanaro et al.[12]. Our work builds on the promising results of Weitz et al.[18] on the max-cut problem by expanding PAOA into a scalable, hardware-compatible, and general-purpose optimization algorithm.

The remainder of this paper is organized as follows: we first present the theoretical formulation of PAOA, including its parameterization, Markov structure, and sampling-based approximation with p-bits. In order to illustrate the correspondence between analytical gradients and derivative-free optimization, we study a simple majority gate problem. We then demonstrate the recovery of SA as a special case and implement this regime on hardware, followed by a description of the FPGA architecture used to accelerate sampling. Using matched parameter counts, we benchmark PAOA against QAOA on the SK model. Finally, we study a heavy-tailed SK variant to show how PAOA can discover multiple annealing schedules that outperform standard single-schedule SA.

## Results and discussion
### Probabilistic Approximate Optimization Algorithm overview
The power of the PAOA comes from a conceptual departure from traditional optimization methods, such as SA. While SA explores a single, fixed energy landscape defined by a model Hamiltonian, PAOA treats the landscape itself as a variational object to be optimized. The goal is to dynamically reshape this landscape to make the ground state of the original problem easier to find.

This creates a two-level optimization loop, as illustrated in Fig. 1. In an outer loop, a classical optimizer adjusts a set of variational parameters, $\boldsymbol{\theta}$. In an inner loop, a probabilistic computer samples states from the variational landscape currently parameterized by $\boldsymbol{\theta}$. In the most general case, $\boldsymbol{\theta}$ specifies the full set of couplings and fields $\{J^{(k)}, h^{(k)}\}$ at each layer $k$, so that the landscape itself is directly reconfigured by the optimizer. Simpler ansätze, such as global or node-wise schedules introduced in the Spectrum of variational ansätze subsection, can be viewed as constrained subsets of this general $J$, $h$ parameterization (for example, a global inverse temperature amounts to scaling all entries of $J$ by a single parameter). The sampler acts on $M$ spins in the ansatz, while the original problem has $N$ visible spins. In general $M \geq N$ because hidden spins may be introduced to enrich the representation, in direct analogy to neural quantum states[24]. Unless otherwise stated, we set $M=N$ in the remainder of the paper, for simplicity, and leave the possibility of introducing hidden spins for better representation to future work.

During training, samples generated under $\boldsymbol{\theta}$ are evaluated with a task-dependent cost function $\mathcal{L}(\boldsymbol{\theta})$. In problems with a known target set of states (e.g., majority logic gate, discussed in the Representative example: majority gate subsection), $\mathcal{L}$ is typically chosen as a negative log-likelihood. In optimization problems such as 3D spin glasses, $\mathcal{L}$ is instead taken as the average energy over sampled configurations. In either case, feedback from the samples updates $\boldsymbol{\theta}$, decoupling the problem energy from the sampling dynamics and enabling PAOA to discover nonequilibrium pathways to solutions.

### Spectrum of variational ansätze
The power and flexibility of PAOA lie in the choice of the variational parameters $\boldsymbol{\theta}$ that define the search landscape. We refer to this parameterization as the variational ansatz. Let $N$ denote the total number of spins in the problem. To explore the tradeoffs between expressiveness and generalization, in this paper, we study four representative ansätze,

defined by the number of free parameters, $\Gamma$:

$$\Gamma \in \left\{1, 2, N, \frac{N(N-1)}{2}\right\}$$

- When $\Gamma = 1$, the landscape is scaled by a single, learned inverse temperature $\beta^{(k)}$ at each layer $k$. This is the global annealing schedule, which is the closest analog to SA, with the crucial difference that the schedule is learned rather than predefined.
- When $\Gamma = 2$, two independent schedules are assigned to distinct parts of the graph, typically divided based on node properties like weighted degree.
- When $\Gamma = N$, each node is assigned a local schedule $\beta_i^{(k)}$, allowing the algorithm to anneal different parts of the problem at different rates.
- When $\Gamma = N(N-1)/2$, the entire symmetric coupling matrix $J_{ij}^{(k)}$ is used as the variational ansatz, giving the algorithm maximum freedom to reshape the landscape at each layer.

Recent work in QAOA has shown that increasing the number of variational parameters per layer can improve convergence and reduce the required circuit depth[8,9]. A similar principle applies to PAOA: more expressive schedules allow the system to reach useful non-equilibrium distributions in fewer layers (e.g., a single layer for a fully parameterized AND gate, as we show in Supplementary Figs. S2, 3), but may also increase the risk of overfitting. The balance between expressiveness and generalization remains a key consideration when selecting an ansatz.

## Implementation via probabilistic computers

The practical implementation of the sampling inner loop is achieved using a probabilistic computing (p-computing) framework[19]. This framework uses networks of binary stochastic units, or p-bits, that sample from Boltzmann-like distributions via asynchronous updates governed by Glauber dynamics. A single p-bit updates its state $m_i$ according to

$$m_i = \mathrm{sgn}\left[\tanh(\beta I_i) - \mathrm{rand}_u(-1, 1)\right] \tag{1}$$

where $\mathrm{rand}_u(-1, 1)$ is a random variable uniformly distributed in the range $[-1, 1]$, $\beta$ is the inverse temperature from the variational ansatz, and the local input $I_i$ is

$$I_i = \sum_j J_{ij} m_j + h_i \tag{2}$$

As long as connected p-bits are updated sequentially, in any random order, this update rule generates samples from a Boltzmann distribution $P(\{m\}) \propto \exp[-\beta E(\{m\})]$ over time, associated with the energy

$$E(\{m\}) = -\sum_{i<j} J_{ij} m_i m_j - \sum_i h_i m_i \tag{3}$$

## Formalism and connection to Markov chains

While in practice, PAOA relies on direct MCMC sampling, the process has a rigorous foundation in the theory of Markov chains. The evolution of the probability distribution $\rho$ over the state space can be described by a sequence of transition matrices. To formalize this, we consider a $p$-layer process where the distribution after $k$ layers, $\rho_k$, is given by

$$\rho_k = W^{(k)}(\boldsymbol{\theta}^{(k)}) \cdots W^{(1)}(\boldsymbol{\theta}^{(1)}) \rho_0 \tag{4}$$

where $\rho_0 \in \mathbb{R}^{2^N}$ is the initial distribution and each $W^{(k)}$ is a $2^N \times 2^N$ transition matrix parameterized by the variational parameters $\boldsymbol{\theta}^{(k)}$ for that layer. For a sequential update scheme, each $W^{(k)}$ can be factorized

as a product of single-site update matrices, $W^{(k)} = w_N^{(k)} w_{N-1}^{(k)} \cdots w_1^{(k)}$. The exact construction of these matrices from the p-bit update rule is detailed in Supplementary Section 3.

This layered application of stochastic matrices to a probability vector $\rho$ is the direct classical analog of applying unitary operators $U^{(k)}$ to a wavefunction $\psi$ in QAOA. The crucial distinction, however, is that each $W^{(k)}$ is a norm-one-preserving stochastic matrix that mixes non-negative probabilities, whereas each $U^{(k)}$ is a norm-two-preserving unitary matrix that rotates complex vectors.

The absence of complex phases in the classical evolution means that PAOA proceeds without the possibility of quantum interference. Consequently, for optimization problems where a constructive interference mechanism for QAOA is unclear, its advantage over classical alternatives is not guaranteed. Indeed, as we will demonstrate in the PAOA versus QAOA: Sherrington-Kirkpatrick model subsection, our benchmarking shows that PAOA consistently reaches higher approximation ratios on the Sherrington-Kirkpatrick model when compared to QAOA using an identical number of variational parameters.

The ultimate objective of PAOA is to find the states $\{m\}$ that minimize the problem's energy function (Eq. (3)). This is achieved through the two-level optimization loop described earlier. The outer loop minimizes a cost function (different from the problem's energy function) over the variational parameters $\boldsymbol{\theta}$, thereby reshaping the sampling landscape to make the optimal states $\{m\}$ more probable and easier to find through MCMC sampling in the inner loop.

For a target set of states $\mathcal{X}$, this cost function can be the negative log-likelihood:

$$\mathcal{L}(\boldsymbol{\theta}) = -\sum_{\{m\} \in \mathcal{X}} \ln\left(\rho_p(\{m\}; \boldsymbol{\theta})\right) \tag{5}$$

In practice, as the exact computation of $\rho_p$ is intractable for large $N$, it is replaced by an empirical distribution $\hat{\rho}_p$ estimated from $N_E$ independent MCMC samples. Because the samples are generated from a shallow Markov process (small $p$), the system typically remains out of equilibrium. This non-equilibrium character distinguishes PAOA from classical annealing methods and may unlock new features such as initial condition dependence and faster convergence to the solution. However, this may also become a liability for shallow circuits, as strong dependence on the initial distribution $\rho_0$ or overfitting may prevent the learned heuristics from generalizing. Therefore, the same $\rho_0$ used during training should also be used during inference. Deep PAOA circuits do not exhibit any initial condition dependence, especially with a small initial $\beta$ that randomizes spins (see the Discovering SA with PAOA subsection).

Finally, the PAOA framework is general in two key respects. First, the cost function being minimized is not restricted to the two-local form, such as the Ising energy of Eq. (3); it can be any computable function over the states $\{m\}$, including higher-order $k$-local Hamiltonians or likelihood functions for machine learning tasks. Second, the variational landscape used for sampling, while implemented here with an Ising-like p-computer, is also not fundamentally limited. The PAOA approach is compatible with any model from which one can efficiently draw samples via MCMC, such as Potts models.

## Representative example: majority gate

To illustrate the behavior of PAOA under both exact and approximate dynamics, we solve a small optimization problem involving a four-node majority gate. This problem serves as a tractable testbed for understanding the role of the classical optimizer.

The majority gate is defined over four binary variables $[A, B, C, Y]$, where the output $Y$ is given by

$$Y = \mathrm{MAJ}(A, B, C) = \begin{cases} A \vee B, & \text{if } C = 1 \\ A \wedge B, & \text{if } C = 0 \end{cases} \tag{6}$$

The eight correct input-output combinations, shown in the truth table in Fig. 2a, define the target set of states $\mathcal{X}$. The goal of the optimization is to find variational parameters that cause the final distribution, $\boldsymbol{\rho}_p$, to be concentrated on this set.

As a starting graph, we consider a fully connected Ising graph with $J_{ij} = +1$ and no biases. In this example, the variational parameters are node-specific schedules $\beta_i(p)$, corresponding to $\Gamma = N$. To validate the role of the numerical optimizer, we compare two optimization strategies: a full Markov chain with gradient-based optimization and an MCMC-based approximation using the COBYLA algorithm[25].

For the gradient-based approach, we use the formulation laid out in the Formalism and connection to Markov chains subsection and optimize over two layers ($p = 2$) with a uniform initial distribution $\boldsymbol{\rho}_0 = 1/16$. The cost is the negative log-likelihood over the eight correct states, and the variational parameters $\beta_i(p)$ are updated using gradient descent with a fixed learning rate $\eta = 0.004$. Optimization stops when either the gradient norm falls below $10^{-7}$ or a fixed iteration budget is reached.

In the MCMC-based approach, the same initial condition is used, but sampling is performed using $10^7$ independent runs. COBYLA is used to update the parameters based on the empirical distribution $\hat{\boldsymbol{\rho}}$. The optimization stops based on parameter convergence or a maximum number of function evaluations. Results are shown in Fig. 2b, where the two curves converge to nearly identical minima.

Figure 2c and d show the final distributions under both methods. The agreement between the exact Markov flow and the sampled histogram confirms that derivative-free optimization closely tracks the true gradient.

**Algorithm 1.** PAOA: global annealing schedule

> **Input** : number of nodes $N$, number of layers $p$, number of
>  sweeps per layer $s/p$, number of experiments $N_E$,
>  problem weight matrix $J$, initial variational
>  parameters $\beta$, variational parameters tolerance $\varepsilon_{\text{step}}$,
>  maximum iterations $t_{\max}$
> **Output**: trained annealing schedule $\beta_{opt}$

1  **Function** p-computer(FPGA) ($\beta, J, N, p, s/p, N_E$):
2 **for** $i \leftarrow 1$ **to** $N_E$ **do**
3 initialize all spins randomly
4 **for** $j \leftarrow 1$ **to** $p$ **do**
5 $\beta \leftarrow \beta(i)$
6 **for** $k \leftarrow 1$ **to** $s/p$ **do**
7 **for** $l \leftarrow 1$ **to** $N$ **do**
8 solve equations (1) and (2)
9 store p-bit states in the BRAM
10 **return** *all stored p-bit states*

11 **Function** PAOA-circuit($\beta, J, N, p, s/p, N_E$):
12 states $\leftarrow$ p-computer(FPGA) ($\beta, J, N, p, s/p, N_E$)
13 compute the energy using equation (3) for all states
14 compute the average energy
15 **return** *average energy*

16 **while** *(step size $> \varepsilon_{step}$ and number of iterations $< t_{\max}$)* **do**
17 average energy $\leftarrow$ PAOA-circuit ($\beta, J, N, p, s/p, N_E$)
18 minimize average energy and get a perturbation vector
 (**p**) using a gradient-free optimizer
19 **for** $i \leftarrow 1$ **to** $p$ **do**
20 $\beta(i)^{t+1} \leftarrow \beta(i)^t + \mathbf{p}(i)$
21 $t \leftarrow t + 1$
22 **return** *optimal variational parameters*

We also test the same procedure on a 5-node full-adder circuit using the fully-parameterized $J_{ij}(p)$ ansatz. The resulting distributions

(see Supplementary Fig. S1) once again demonstrate tight correspondence between exact and approximate optimization at each layer.

Although analytical optimization is feasible for small $N$, the $2^N \times 2^N$ transition matrix becomes intractable for larger systems. The success of COBYLA in this setting confirms that PAOA can be implemented efficiently on general p-computers using standard MCMC.

Interestingly, PAOA can explore parameter regimes not accessible to traditional Ising formulations. For example, optimal $\beta$ values may be negative, corresponding to negative temperatures without a clear interpretation in the statistical physics-based context. Since equilibrium Boltzmann-sampling is not necessarily the starting point, alternative and more hardware-friendly activation functions could also be explored with PAOA, such as replacing the hyperbolic tangent in Eq. (1) with the error function erf, or with saturating linear functions.

The shallow nature of the Markov chain used in PAOA introduces a dependency on the initial distribution $\boldsymbol{\rho}_0$. In contrast to standard Boltzmann training methods such as Contrastive Divergence[26], which seek to approximate the equilibrium distribution through extensive sampling, PAOA optimizes the evolution of the distribution under a fixed initialization. This shortcut avoids the computational burden of long mixing times but restricts the learned dynamics to a specific initialization at shallow depth. While this dependence can hinder generalization at shallow depth, as we demonstrate in the Discovering SA with PAOA subsection, for deep PAOA circuits, initialization dependence is not an important concern.

## Discovering simulated annealing with PAOA

An intriguing question is whether PAOA, when restricted to a global annealing schedule with one parameter per time step, can recover the well-known structure of SA. In this section, we show that the answer is yes. Using a minimal parameterization with a single global inverse temperature, $\beta^{(k)}$, for each one of the $p$ total layers, PAOA is able to discover SA-like profiles that optimize the average energy on a large three-dimensional (3D) spin-glass instance.

We apply PAOA to a 3D cubic lattice of size $L^3 = 6^3$, which yields a sparse bipartite interaction graph. To leverage this structure for massive parallelism on our FPGA-based p-computer, we use chromatic updates[21,22]. We employ the simplest $\Gamma = 1$ global annealing ansatz, where all nodes share the same scalar inverse temperature $\beta^{(k)}$ at each layer $k$, forming a schedule of total length $p \in \{5, 10, 15\}$.

Sampling is performed on hardware using 720 Monte Carlo sweeps (MCS) per layer. We choose a uniform initial distribution, achieved by appending $\beta = 0$ to the beginning of the schedule. After sampling, configurations are transferred to the CPU, where the CPU evaluates the average energy $\langle E \rangle$ using Eq. (3), and COBYLA updates the schedule $\beta^{(k)}$. In this training loop, the cost function is the average sampled energy, $\langle E \rangle$, estimated from $N_E$ independent MCMC runs. After training, however, each candidate's schedule is evaluated on a fresh batch of $10^5$ runs and assigned a success probability, defined as the fraction of runs that reach the putative ground-state energy at the final layer. When referring to the best schedule or best $\beta$ in Fig. 3, we mean the schedule that achieves the highest success probability out of this independent batch. Insets in Fig. 3c show these success probabilities for all 100 optimizer runs, illustrating both the variability across runs and the improvement with circuit depth.

The loop continues until either convergence or a maximum number of energy evaluations is reached. The algorithm is outlined in Algorithm 1, and parameters are listed in Table 1.

The results of this optimization are summarized in Fig. 3. The optimized schedules shift the energy distribution toward lower values (Fig. 3b), increasing the probability of reaching the putative ground state at $E = -360$, which was independently found using a linear SA (from $\beta = 0.1$–5) schedule with $10^6$ MCS. The schedules responsible for this improvement are shown in Fig. 3c. Notably, without any constraints on monotonicity, PAOA consistently learns schedules that

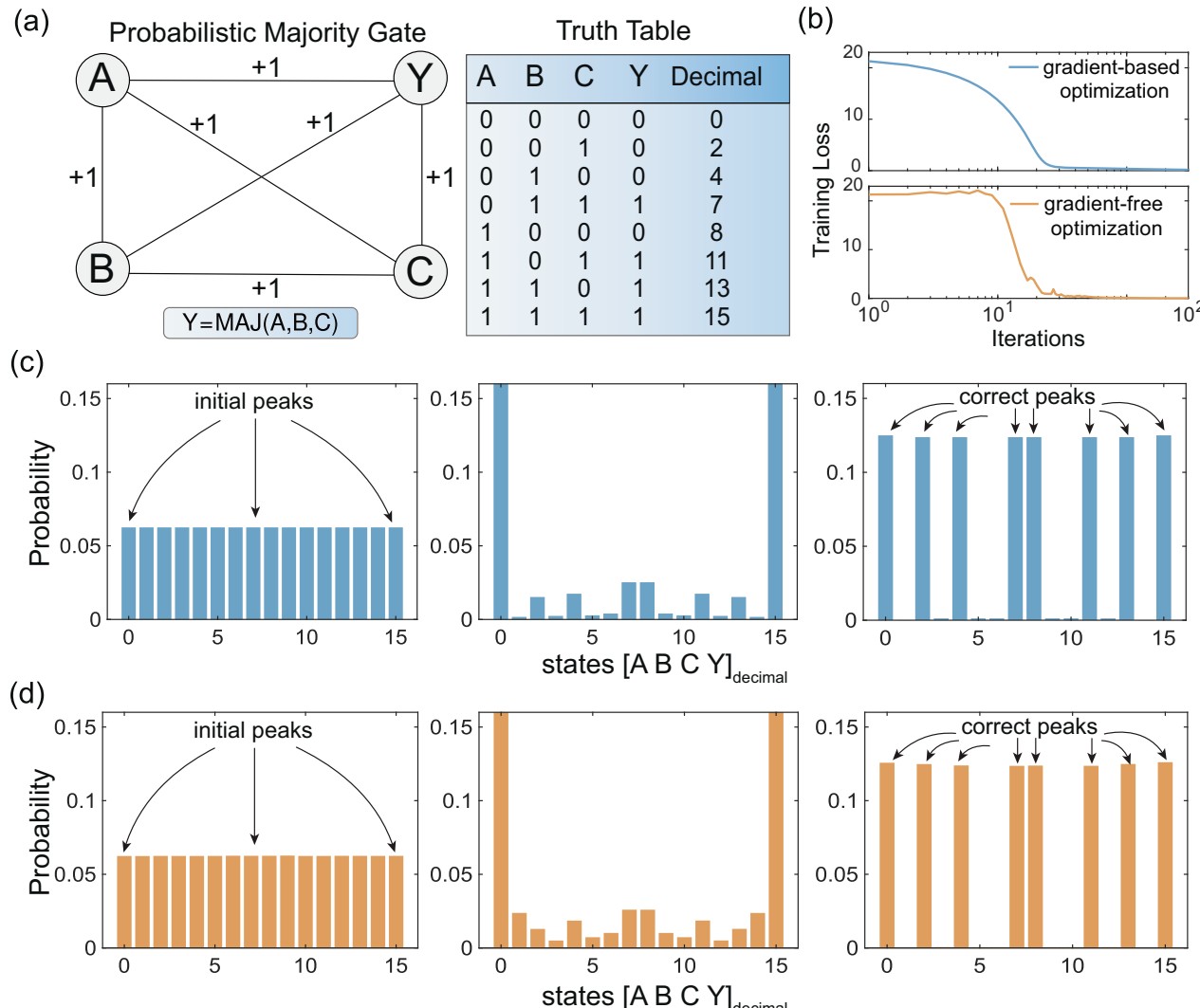

**Fig. 2 | Majority gate benchmark: comparing analytical and sampling-based PAOA. a** Fully connected four-node network used to implement the majority gate $Y = \mathrm{MAJ}(A, B, C)$, where $Y = A \wedge B$ if $C = 1$ and $Y = A \vee B$ if $C = 0$. The table lists the eight valid input-output combinations, labeled by their decimal encoding. **b** Training loss during optimization, comparing exact gradients (blue) to gradient-free optimization using COBYLA (orange) with $10^7$ MCMC samples. Both methods converge to the same minimum. **c** Time evolution of the exact distribution over two layers using analytical Markov matrices. Initial uniform distribution ($p = 0$) is transformed into a peaked distribution ($p = 2$) concentrated on the correct truth table entries. **d** Corresponding evolution using MCMC samples and COBYLA. The approximated distributions closely match the exact dynamics.

begin at high temperature (low $\beta$) and gradually cool (increase $\beta$), resembling classical SA despite starting from a flat $\beta = 2$ schedule.

As depth $p$ increases, our schedules both cool further and spend more total sweeps at lower temperatures, so the energy histograms naturally put more weight near the ground state. In majorization language, the deeper runs appear to majorize the shallower ones, but we only show energy marginals; therefore, this is suggestive rather than proof. Similar cooling-with-depth trends were observed for QAOA[27].

The elevated $\beta$ at the first step ($p = 1$) for the 15-layer case may be a result of the substantial degeneracy among near-optimal schedules. Many distinct $\beta$ profiles achieve nearly identical success probabilities (shaded blue lines in Fig. 3c). The inset histograms show that while individual schedules vary across 100 optimizer runs, performance remains stable (even for non-monotonic schedules) with the smallest variance in success probabilities for deeper schedules ($p = 15$).

Constraints could be added to restrict the class of allowed schedules, e.g., enforcing linear, exponential, or monotonic cooling profiles using inequality constraints in COBYLA. We did not use such constraints to avoid biasing the solution toward SA. These results demonstrate that SA emerges as a special case of variational Monte Carlo, recoverable from the global schedule ansatz. Moreover, PAOA proves to be more general, as it points to the possibility of alternative non-monotonic schedules with comparable performance, indicating the flexibility of the algorithm.

## PAOA on FPGA with on-chip annealing

To support long annealing schedules in 3D spin-glass problems, we designed a p-computer architecture that performs annealing entirely on-chip, unlike earlier implementations that require off-chip resources for annealing (e.g., ref. 21). The annealing schedule $\beta^{(k)}$ is preloaded to the FPGA and indexed via a layer counter that advances after a fixed number of MCS. At each step, the current $\beta$ value is used to scale the synaptic input $I_i$ via a digital signal processing multiplier (see Supplementary Fig. S4).

The full $p$-layer annealing process runs uninterrupted on the FPGA, and the final spin configurations are read out only after the last layer is complete.

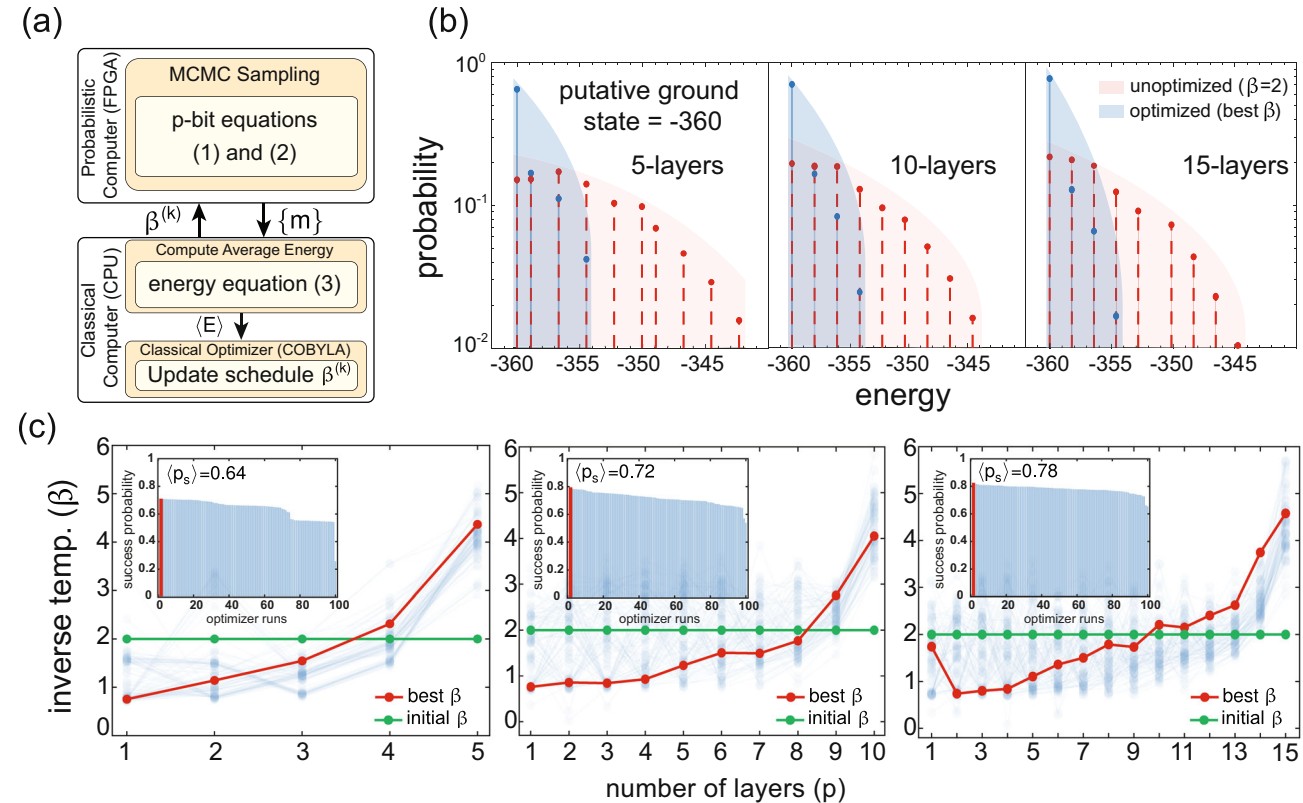

**Fig. 3 | Discovering simulated annealing with PAOA on a 3D spin-glass problem.**
**a** The hybrid architecture combines an FPGA-based p-computer for MCMC sampling, where {m} represents the sampled state, with a classical CPU that optimizes the global annealing schedule ($\beta^{(k)}$) using the average energy ($\langle E \rangle$) computed across independent experiments/runs. **b** Energy histograms on a single $L^3 = 6^3$ instance before ($\beta = 2$, red) and after (best $\beta$ schedule, blue) optimization. The optimized schedule shifts the distribution (shaded area) toward the putative ground state

(energy = −360), increasing its discovery frequency out of $N_E = 10^5$ independent runs. **c** Optimized schedules for $p$-layer architectures where $p \in \{5, 10, 15\}$. Starting from a flat initial schedule ($\beta = 2$, green), PAOA consistently discovers cooling schedules (best shown in red) that resemble SA. Faint curves show all 100 optimization runs. The insets display the sorted success probabilities, demonstrating that deeper architectures improve the average success probability ($\langle p_s \rangle$) and reduce run-to-run variability.

### Table 1 | Parameters used in Algorithm 1

| Parameter | Value |
|---|---|
| number of nodes ($N$) | 216 |
| number of Monte Carlo Sweeps (MCS)/layer ($s/p$) | 720 |
| number of layers ($p$) | {5, 10, 15} |
| initial variational parameters ($\beta(p)$) | 2 |
| variational parameters tolerance ($\varepsilon_{step}$) | $10^{-4}$ |
| maximum iterations ($t_{max}$) | 5000 |
| number of independent experiments ($N_E$) | $10^5$ |

The FPGA operates in fixed-point arithmetic with $s\{4\}\{5\}$ precision (where $s$ denotes the sign bit, followed by 4 integer and 5 fractional bits) for both $\beta^{(k)}$ and $J_{ij}$. Due to the size of the FPGA we use, we instantiate 10 independent replicas of the $L^3 = 6^3$ spin-glass graph, enabling parallel updates of 2160 p-bits per cycle. Our FPGA architecture with on-chip annealing capability achieves about approximately an 800-fold reduction in wall-clock time in the FPGA compared to an optimized CPU implementation with graph-colored Gibbs sampling (Table 2).

PAOA is parallelizable at multiple levels. First, the cost estimator $\langle E \rangle$ is computed from $N_E$ independent MCMC runs, which can be executed concurrently across replicas. Second, on sparse graphs, independent sets admit parallelism via chromatic sampling. Consequently, high-throughput implementations are possible using CPUs, GPUs, FPGAs, custom accelerators or nanodevices. Our FPGA design is one point in this space and was chosen to accelerate the experiments

### Table 2 | Wall-clock time for 3D spin glass ($L^3 = 6^3$), comparing CPU to FPGA with 10 identical replicas and $10^4$ independent runs

| Layers | 5 | 10 | 15 | Flips/ns |
|---|---|---|---|---|
| Time (MCS/replica) | $3.6 \times 10^7$ | $7.2 \times 10^7$ | $1.08 \times 10^8$ | |
| CPU (s) | 1953 | 3800 | 5850 | 0.0398 |
| FPGA-10 replicas (s) | 2.46 | 4.90 | 7.36 | 31.74 |

in this work. The wall-clock numbers in Table 2 are an implementation case study rather than a universal hardware comparison.

All simulations use a uniform initialization, implemented by prepending a $\beta = 0$ layer to the schedule. This yields random initial conditions and ensures randomized spin states for each replica. Further hardware design details are provided in Supplementary Section 4.

### PAOA versus QAOA: Sherrington-Kirkpatrick model

The Sherrington-Kirkpatrick (SK) model defines a mean-field spin glass with all-to-all random couplings, and serves as a canonical benchmark for optimization algorithms. The classical energy function is defined as

$$E(\{m\}) = -\frac{1}{\sqrt{N}} \sum_{i<j} J_{ij} m_i m_j \tag{7}$$

where $m_i \in \{-1, 1\}$ and $J_{ij} \sim \mathcal{N}(0, 1)$. The exact ground-state energy per spin in the large-$N$ limit is known[28].

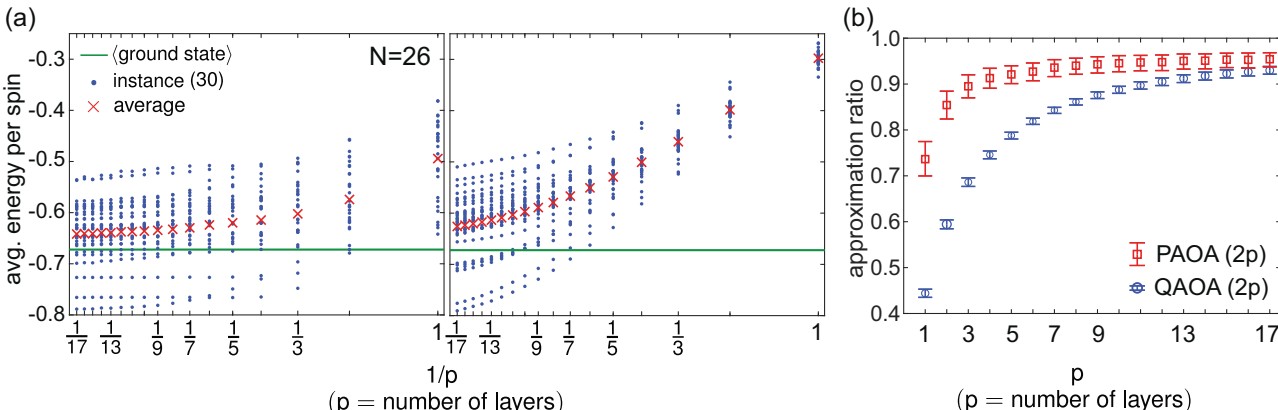

**Fig. 4 | PAOA vs QAOA on the Sherrington-Kirkpatrick model. a** PAOA results (left) using two-schedule ansatz ($\beta_1$ and $\beta_2$) with $2p$ parameters compared against QAOA (right) with $2p$ parameters ($\gamma$ and $\beta$). For each depth $p$, the PAOA schedules are optimized on a separate training set; the average schedule is then applied to 30 random test instances of size $N = 26$ without retraining. QAOA results use optimal parameters from prior work[7,29]. Red crosses denote averages across the 30 instances, blue dots show individual instance energies, and the solid green line indicates the average ground-state energy per spin. **b** Approximation ratios of PAOA (red squares) and QAOA (blue circles), averaged across the 30 instances. Error bars indicate the 95% confidence intervals computed from $10^4$ bootstrap samples with replacement.

QAOA has been studied extensively on the SK model[7,15,29,30], with recent numerical work suggesting it can approximate the ground state in the infinite-size limit[31]. To enable a direct comparison, we evaluate PAOA using a two-schedule ansatz with exactly $2p$ parameters, matching the parameter count of depth-$p$ QAOA. Although PAOA scales readily to much larger $N$, we stick to a fixed size of $N = 26$, a typically studied SK-model, to enable a direct comparison with QAOA.

Each schedule is assigned to one half of the graph, split uniformly at random. For $N = 26$, we randomly generate 30 training instances and optimize PAOA separately for each instance and layer depth $p \in \{1, ..., 17\}$. The resulting schedules are then averaged and applied without further adjustment to a disjoint set of 30 test instances. QAOA results plotted in Fig. 4a use optimal parameters from prior work[7,29]. For each instance, exact ground states are computed by exhaustive search.

Our results show that PAOA has highly competitive performance under these iso-parametric conditions. As shown in Fig. 4a (left, PAOA), the average energy per spin consistently decreases with layer depth, and schedules trained on $N = 26$ generalize well to much larger instances up to $N = 500$ (see Supplementary Fig. S5). To formalize this comparison, we report the approximation ratio in Fig. 4b, defined as

$$\text{Approx. Ratio} = \frac{E(\{m\})}{E(\{m_{\text{sol}}\})} \tag{8}$$

where $E(\{m_{\text{sol}}\})$ is the cost of the exact ground state. PAOA achieves a consistently higher approximation ratio than standard QAOA across all depths. Notably, the performance of vanilla PAOA is comparable to that of recently proposed hybrid quantum-classical methods that use classical techniques to improve QAOA's performance[32], suggesting that a definitive quantum advantage on this problem has yet to be firmly established.

We emphasize that a clear quantum advantage over classical algorithms is expected to emerge in problems where interference plays an explicit role, such as in the synthetic examples constructed by Montanaro et al.[12]. In these cases, the superposition of many computational paths, enabled by either the problem structure or the quantum algorithm, creates interference patterns that amplify the correct solution while suppressing others, an effect that classical probabilistic samplers struggle to reproduce due to the well-known "sign problem"[33]. As shown in ref. 34, a probabilistic formulation of generic unitary evolutions yields complex (often purely imaginary) weights, so sampling effectively becomes randomized. While post-phase corrections recover unbiased estimates, the number of samples grows exponentially.

These interference-dominated regimes are precisely where we do not expect PAOA, or any classical sampler, to compete. On the other hand, unless such interference-driven advantages are clearly identified and demonstrated more broadly across practical problem classes, we do not expect any significant quantum advantage from QAOA. Meanwhile, PAOA offers a rigorous, scalable and high-performing benchmark for variational sampling at problem sizes beyond the reach of current quantum devices.

## Multi-Schedule annealing in the Lévy SK model

To a heuristic solver, the SK model presents no intrinsic structure beyond a dense random coupling graph. To explore whether PAOA can adaptively exploit structure when it might exist, we consider a variant where the coupling weights $J_{ij}$ follow a heavy-tailed Lévy distribution[35]:

$$P(J) = \frac{\alpha}{2} |J|^{-1-\alpha}, \quad |J| > 1 \tag{9}$$

We use $\alpha = 0.9$, which produces a distribution with diverging variance, dominated by rare, large couplings.

In this regime, a natural question arises: can PAOA discover principles that treat strongly and weakly coupled nodes differently? Prior work on Boltzmann machines and annealing schedules[36,37] suggests that high-degree or strongly coupled nodes benefit from slower annealing (more time for high temperature) to avoid freezing too early. In these prior works, the heuristics are hand-designed by the ingenuity of the human algorithm designers. Here, we test whether the principle of annealing subgraphs with different temperatures can be discovered by PAOA through black-box optimization.

We begin by introducing a coupling strength per node:

$$\lambda_i = \sum_j |J_{ij}| \tag{10}$$

We then sort nodes by $\lambda_i$ and split into a heavy group (top 20%) and a light group (bottom 80%), as shown in Fig. 5a. The split here is arbitrary and we confirmed that a 50–50% partition following the same procedure yields similar results (see Supplementary Fig. S6).

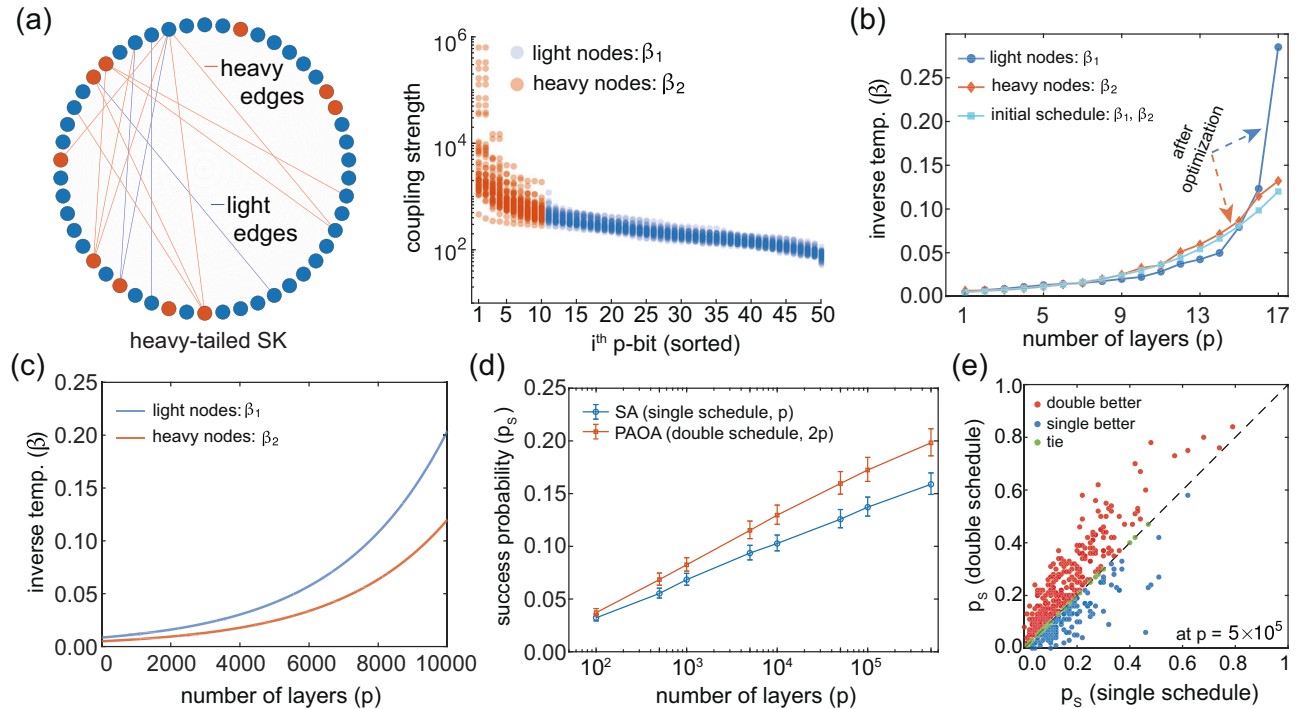

**Fig. 5 | Learning a variational principle using a PAOA double-schedule ansatz.**
**a** Heavy-tailed SK: per-node coupling strengths for 50 instances of $N = 50$, sorted in descending order; the heavy-tailed distribution separates heavy (assigned $\beta_2$ schedule) and light (assigned $\beta_1$ schedule) nodes. **b** PAOA training with two schedules ($2p$ parameters) assigned to heavy and light nodes based on their coupling strengths, initialized from an optimized single annealing schedule (cyan). Curves are averaged across 50 instances; red/blue denote heavy/light nodes. **c** The extrapolated double schedule suggested by **b**: the heavy-node schedule is extended to more layers, and the light-node schedule is scaled up following PAOA's guidance. **d** Average success probability over 500 instances comparing single-schedule SA (blue) and double-schedule PAOA (red) for $N = 50$. Error bars are 95% confidence intervals from $10^5$ bootstrap samples with replacement. **e** Success probability showing per-instance comparison between PAOA ($2p$) and SA ($p$) at $p = 5 \times 10^5$.

To test if PAOA can leverage this structure, we first identify a near-optimal single-schedule using SA at shallow depth ($p = 17$). We then let COBYLA optimize two-schedules, corresponding to heavy and light nodes, freely. After convergence, we average the schedules across all instances. As shown in Fig. 5b, the optimized schedules exhibit a qualitative separation: heavy nodes are assigned a higher-temperature (lower $\beta$) profile, and light nodes anneal faster. We verified that the different schedules for heavy and light nodes lead to higher success probability for the double-schedule over the single-schedule. Due to the shallow depth ($p = 17$) however the differences are not appreciably large.

However, when extrapolating the double-schedules discovered by PAOA to much deeper layers (Fig. 5c), we observe a clear difference against single-schedule SA. Our extrapolation is relatively straightforward but we report the details in the Methods Section. The results, shown in Fig. 5d for $N = 50$ and 500 unseen instances, show that the two-schedule ansatz outperforms the single schedule, with a clear gap in success probabilities between the two approaches, confirming a robust benefit from structured, heterogeneous annealing. Details on determining the success probability for single and double schedules are in the Methods Section.

This behavior suggests that PAOA can discover problem-aware heuristics computationally. While such adaptive schedules have long been used manually in structured settings[38], we are not aware of prior variational algorithms for non-equilibrium sampling that systematically learn them from data. In this sense, PAOA serves as a tool for discovering new algorithmic strategies in disordered systems.

## Outlook
We have introduced the PAOA, a variational Monte Carlo framework that generalizes SA and supports a wide range of parameterizations compatible with existing Ising hardware. By treating the energy landscape itself as a variational object and updating it through classical feedback from Monte Carlo samples, PAOA enables efficient sampling from non-equilibrium distributions using shallow Markov chains.

We demonstrated that PAOA captures SA as a limiting case and, without constraints, can rediscover SA-like behavior when optimized for energy minimization. Using a minimal global schedule ansatz, PAOA efficiently learned monotonic cooling profiles that converge to low-energy states on 3D spin-glass problems. The approach was implemented on an FPGA-based p-computer, achieving hardware-accelerated annealing at rates exceeding CPU-based implementations by several orders of magnitude.

Beyond global schedules, we explored richer ansätze with multiple temperature profiles. On the Sherrington-Kirkpatrick (SK) model, PAOA exceeded the performance of QAOA under iso-parametric conditions, while scaling to system sizes beyond those accessible to quantum devices. In a heavy-tailed variant of the SK model, PAOA automatically learned to assign slower annealing schedules to strongly coupled nodes, reproducing hand-crafted heuristics from prior work. As such, PAOA can be a tool for discovering new algorithmic strategies for sampling in disordered systems.

## Methods
Numerical simulation uses double precision (64-bit) in C++ to generate results for PAOA in Figs. 2d and 4a, and Supplementary Fig. S1d. QAOA results were reproduced from ref. 39.

### Data transfer between FPGA and CPU
A PCIe interface was used to communicate between FPGA and CPU through a MATLAB interface for the "read/write" operations. A global "disable/enable" signal broadcast from MATLAB to the FPGA was used

to freeze and resume all p-bits. Before a "read" instruction, the p-bit states were saved to the local block memory (BRAM) with a snapshot signal. Then the data were read once from the BRAM using the PCIe interface and sent to MATLAB for post-processing, that is, updating the annealing schedule ($\beta^{(k)}$). For the "write" instruction, the "disable" signal was sent from MATLAB to freeze the p-bits before sending the updated schedule. After the "write" instruction was given, p-bits were enabled again with the "enable" signal sent from MATLAB.

### Measurement of wall-clock time and flips per nanosecond

To measure the annealing wall-clock time per replica in FPGA along with the reading and writing overhead, we use MATLAB's built-in tic and toc functions and average the total annealing time over 100 independent experiments. Adding all flip attempts for the entire system with all replicas, 2160 p-bits, the total flips per nanosecond (flips/ns) are computed. For CPU measurements, MATLAB's built-in tic and toc functions were used to measure the total annealing time taken to perform the MCS in Table 2 with 1000 runs each. The average time per MCS is reported for a single replica for a network size of $L^3 = 6^3$.

### Schedule generation for the Lévy SK model benchmark

The performance benchmark in the Multi-Schedule annealing in the Lévy SK model subsection required comparing two annealing strategies. The schedules for these strategies were generated via a grid search based on a geometric functional form:

$$\beta^{(k)} = \beta_{\text{initial}} \cdot \left( \frac{\beta_{\text{final}}}{\beta_{\text{initial}}} \right)^{k/(p-1)}, \; k = 0, \ldots, p-1 \qquad (11)$$

The parameter ranges searched were $\beta \in [0.005, 0.12]$ for the $N = 50$ instances.

The two strategies were constructed as follows:

- Single-Schedule (SA): We performed a grid search over $\beta_{\text{final}}$ to find the single geometric schedule that produced the lowest average energy across all instances. This served as the baseline.
- Two-Schedule PAOA: The optimized baseline schedule was assigned to the heavy nodes, as suggested by PAOA. For light nodes, we scale up the baseline schedule by a scaling factor that reflects the separation proposed by PAOA. Then, both geometric schedules are extrapolated to large number of layers to study the effect of deep PAOA.

In order to compare the two approaches, we define success probability as the ratio of the number of runs that hit the putative ground state to the total number of runs (i.e., 100 independent runs). The putative ground state was found by running a long SA with one million sweeps and ten independent runs, then choosing the lowest energy among all ten million states.

### Optimizer choice

For the outer optimization loop, we used the COBYLA algorithm as it is a robust and widely used method for derivative-free optimization problems with a moderate number of variables[40–42]. Our COBYLA implementation is based on the non-linear optimization package[43]. While the performance of PAOA may depend on the choice of classical optimizer, a detailed comparison of different optimization techniques is beyond the scope of this work.

## Data availability

All generated and processed data used in this study are openly accessible and can be found in the GitHub repository[44]. Other findings of this study are available from the corresponding authors upon request.

## Code availability

The MATLAB and C++ implementations of PAOA used to generate the results and plots in this study can be found in the GitHub repository mentioned in the Data availability section.

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

## Acknowledgements

A.S.A., S.C., and K.Y.C. acknowledge support from the National Science Foundation (NSF) under award number 2311295, and the Office of Naval Research (ONR), Multidisciplinary University Research Initiative (MURI) under Grant No. N000142312708. F.M. was partially supported by the Office of Naval Research (ONR) under Award No. N00014-23-1-2771. We are grateful to Navid Anjum Aadit for discussions related to the hardware implementation of online annealing, and Ruslan Shaydulin and Zichang He for input on QAOA benchmarking. Use was made of Computational facilities purchased with funds from the National Science Foundation (CNS-1725797) and administered by the Center for Scientific Computing (CSC). The CSC is supported by the California NanoSystems Institute and the Materials Research Science and Engineering Center (MRSEC; NSF DMR 2308708) at UC Santa Barbara.

## Author contributions

A.S.A. and K.Y.C. conceived the study. A.S.A. led the full implementation of PAOA, including all simulations and experiments, building on the initial PAOA implementation by S.C. that used black-box optimizers. A.S.A. designed and built the FPGA implementation of PAOA with on-chip annealing capability. S.C. performed the QAOA benchmarking. F.M. contributed key theoretical insights and assisted in interpreting the spin-glass benchmarks. All authors contributed to discussing the results and writing the manuscript.

## Competing interests

The authors declare no competing interests.
