## [Transparent Peer Review file · Nature Communications]

Generalized Probabilistic Approximate Optimization Algorithm

Corresponding Author: Professor Kerem Camsari

Version 0:

Reviewer comments:

Reviewer #1

(Remarks to the Author)

The authors discuss the Probabilistic Approximate Optimization Algorithm (PAOA), which is a classical probabilistic variational circuit inspired by Quantum Approximate Optimization Algorithm (QAOA). PAOA was introduced by Weitz et al in a paper published in PRA earlier this year. The authors implement the algorithm on an FPGA-based probabilistic computer and demonstrated significant speedups over other approaches including QAOA for a small instance of Sherrington–Kirkpatrick model. The authors note that their algorithm can discover annealing schedules that are better than single-schedule simulated annealing.

The authors generalize the original PAOA framework and realize it experimentally with FPGAs. They demonstrate the algorithm on graphs with up to 500 vertices. While this looks impressive, I see several problems with the paper that would need to be sorted out before it is accepted in Nature Communications.

In my opinion, the advantages that the authors claim are not yet fully validated in terms of scalability, generalization and comparison fairness. The fundamental weakness of the approach as compared to simulated annealing is strong dependence on the initial distribution. This can limit the generalization, as the authors admit. Heuristics learned for one initialization may not transfer to others. In the present manuscript, it is not clear to what extent the advantages of PAOA are real or are they the artifacts of carefully chosen initial conditions.

I find the QAOA comparison misleading. PAOA has already been compared to QAOA by Weitz et al, so why the authors repeat similar discussion here? Did the authors find an example that shows QAOA superiority? It would be good to see examples that point towards QAOA advantage as well. The message coming from the PAOA-vs-QAOA section seems to be "PAOA is a practical baseline" – this has been established before.

I find parts of the presentation ambiguous. The authors repeatedly claim that simulated annealing emerges as a limiting case of their approach. Is PAOA then just a generalization of simulated annealing or is there a new class of problems that PAOA can solve better? I am trying to understand whether this approach is fundamentally different or an incremental improvement over old techniques. The paper fails to convincingly establish the approach as a genuinely new paradigm. So far, the presentation and results suggest that the method is an optimized heuristic that may work on carefully chosen benchmarks and only when properly initialized.

(Remarks on code availability)

Reviewer #2

(Remarks to the Author)

Referee Report on Abdelrahman et al. "Probabilistic Approximate Optimization: A New Variational Monte Carlo Algorithm"

The paper introduces the Probabilistic Approximate Optimization Algorithm (PAOA), a new variational Monte Carlo framework. PAOA was inspired by the Quantum Approximate Optimization Algorithm (QAOA) and thus is a "dequantized"

version of QAOA. The authors show that PAOA generalizes simulated annealing and can rediscover annealing schedules. They demonstrate hardware acceleration on an FPGA p bit computer with large performance gains over CPUs. PAOA is benchmarked against QAOA on the Sherrington–Kirkpatrick model and achieves higher approximation ratios under matched parameters. The work is significant because it provides a scalable classical baseline and a tool for discovering new non-equilibrium heuristics, with impact on optimization and probabilistic computing.

I also wanted to point out, in the spirit of scientific reproducibility, the authors have open-sourced their GitHub repository, which is commendable. I took a cursory look at the code.

I think the work is clear, careful, and highly novel. I strongly support publication of this work. However I would like the authors to respond to the comments/clarifications/questions below.

Comments, Clarifications, Questions

Most of my comments are aimed at improving clarity and presentation. The authors should feel free to implement only those changes they believe clarify or strengthen the paper. If a suggested change is marginal, it need not be implemented.

1. Definition of Symbols and Notation

- Define N in the abstract and again in Section II. From my understanding
- N = number of spins/variables in the optimization problem.
- 2^N = size of the full state space.
- In Fig. 1, explicitly define symbols: ρ , $\hat{\rho}$, p , M , m .
- Clarify upfront why M (ansatz size, possibly with hidden variables) differs from N (problem size). This was explained later in Section II.C, but until I read that part I was confused.

2. Section IV and Fig 3

- In Section IV, is "best beta" the optimization criterion that minimization of expected energy, i.e. $\text{best} \equiv \min \langle E \rangle$.
- In the inset figures in Fig 3 (c). State / clarify that $\langle p \rangle$ in plots refers to an expectation over independent runs. That if I was to compute the average of the numbers shown that I'd get the average number.
- Numerical Anomalies? In Fig. 3, the higher energy observed at $p = 1$ for 15 layers is unexplained. Is this just a numerical artifact? It may or may not. Some comment would help the reader.
- I noticed in Fig. 3(b) an apparent majorization ordering across circuit depths. Ordering probabilities from largest to smallest gives $P_{15} \succcurlyeq P_{10} \succcurlyeq P_5$, suggesting structured improvement beyond average energy. Since entropy and purity are Schur-convex/concave, one expects $H(P_{15}) < H(P_{10}) < H(P_5)$. Entropy and purity from truncated data may not show this, but the full distributions likely would. This indicates that adding layers reduces entropy and concentrates probability on the correct answer. This may or may not be an interesting additional insight for the manuscript.

3. The y-axis in Fig. 4.

- The scale seems a little off. Is it logarithmic, or is there a label typo from illustrator editing? The spacing between $|-0.8 - (-0.6)| \neq |-0.6 - (-0.4)|$. Regardless I wonder if clipping the y axis to -0.3 might allow for the data to look less compressed. Or rescaling (possibly log scale) to improve readability. [This also applies to Figure S5]

4. Fig. 5 e.

I'm trying to get a feel for the improvement of PAOA over SA. Are the differences shown meaningful? Or am I missing the point of what is illustrated by the plots here?

5. Reference [31] should be corrected/clarified: Make it clear that it is Chapter 10 in Simulated Annealing, Emile Aarts, Jan Korst, and Wil Michiels.

(Remarks on code availability)

Version 1:

Reviewer comments:

Reviewer #1

(Remarks to the Author)

The authors answered my questions in a satisfactory way. They clarified their claims and improved the presentation of their results. I now support the publication of the manuscript in its present form.

(Remarks on code availability)

Reviewer #2

(Remarks to the Author)

The authors have substantially revised the manuscript and addressed my comments.

I reread the paper and noticed a few minor typos and inconsistencies, that will likely be caught at the proof stage. I did catch one issue. You compute an estimated distribution q_f but never define f . Define it on first use (e.g., $f = \text{"final layer"}$?).

Admittedly I got side tracked by an interesting observation about figure 5 (d). I spent 3 hours exploring that data. Clearly that was research and not related to the report. I hope many other researchers get side tracked by the wonderful results in this manuscript.

I recommend publication without further delay. I am looking forward to reading the published version of the article.

(Remarks on code availability)

Re: NCOMMS-25-58950-T
Generalized Probabilistic Approximate Optimization Algorithm

Abdelrahman S. Abdelrahman,¹ Shuvro Chowdhury,¹ Flaviano Morone,² and Kerem Y. Camsari¹

¹*Department of Electrical and Computer Engineering,
University of California, Santa Barbara, Santa Barbara, CA 93106, USA*

²*Center for Quantum Phenomena, Department of Physics,
New York University, New York, New York 10003 USA*

(Dated: October 10, 2025)

=====

I. REVIEWER #1

=====

The authors discuss the Probabilistic Approximate Optimization Algorithm (PAOA), which is a classical probabilistic variational circuit inspired by Quantum Approximate Optimization Algorithm (QAOA). PAOA was introduced by Weitz et al in a paper published in PRA earlier this year. The authors implement the algorithm on an FPGA-based probabilistic computer and demonstrated significant speedups over other approaches including QAOA for a small instance of Sherrington–Kirkpatrick model. The authors note that their algorithm can discover annealing schedules that are better than single-schedule simulated annealing. The authors generalize the original PAOA framework and realize it experimentally with FPGAs. They demonstrate the algorithm on graphs with up to 500 vertices. While this looks impressive, I see several problems with the paper that would need to be sorted out before it is accepted in Nature Communications. In my opinion, the advantages that the authors claim are not yet fully validated in terms of scalability, generalization and comparison fairness.

AUTHOR RESPONSE

We sincerely thank the reviewer for their thoughtful review and the open-minded assessment. We believe we have addressed each of the points in the revised manuscript, and a color-tracked `diff` file highlighting every change is appended for the reviewer’s convenience.

(1) The fundamental weakness of the approach as compared to simulated annealing is strong dependence on the initial distribution. This can limit the generalization, as the authors admit. Heuristics learned for one initialization may not transfer to others.

AUTHOR RESPONSE

We thank the reviewer for highlighting the role of initialization. Our original text did not clearly separate the behavior of *shallow* versus *deep* PAOA circuits. We would like to make a few clarifying points:

(i) PAOA strictly contains simulated annealing (SA) as a special case (global schedule with one parameter per layer), while also supporting richer, structured parameterizations.

(ii) Initialization dependence is *not* an inherent limitation of PAOA. It only arises in shallow, non-equilibrium settings.

- **Shallow PAOA.** For small depth p , PAOA implements a short Markov flow that can depend on the chosen initialization ρ_0 . This can be advantageous for targeted sampling, e.g., in the majority-gate example (Fig. 2) a specific initial state is steered into a distribution peaked on the correct truth-table entries.
- **Deep PAOA.** With sufficient layers and sweeps, the learned schedules start *hot* (small β), which rapidly mixes the chain and **erases memory of** ρ_0 before cooling. In our 3D spin-glass experiment (Fig. 3), PAOA recovers SA-like behavior *without* imposing monotonicity constraints on β , which is known to be initial condition independent.

(iii) PAOA’s main advantage is expressivity. As an example, in the SK–Lévy model, PAOA discovers a *double-schedule* ansatz that assigns a slower (higher-temperature) annealing schedule to “heavy” nodes and a faster one to “light” nodes. This *learned* heterogeneity outperforms a highly optimized single-schedule SA (revised Fig. 5 and Section VII).

(iv) The learned schedules already **generalize** across instances and sizes **without retraining**. In Fig. 4, schedules trained on one set of $n=26$ SK instances are applied to a disjoint test set; the same schedules also generalize to $n \in \{400, 500\}$ (Fig. S5). Likewise, in Fig. 5, the SK–Lévy heuristics learned at small n are used at larger n without retraining.

AUTHOR ACTION

We revised the second to last paragraph of Section IIB, where we discuss initialization, as follows:

Because the samples are generated from a shallow Markov process (small p), the system typically remains out of equilibrium. This non-equilibrium character distinguishes PAOA from classical annealing methods and may unlock new features such as initial condition dependence and faster convergence to the solution. However, this may also become a liability, for shallow circuits, as strong dependence on the initial distribution ρ_0 or overfitting may prevent the learned heuristics from generalizing. Therefore, the same ρ_0 used during training should also be used during inference. As we show in later Section IV, deep PAOA circuits do not exhibit any initial condition dependence especially with small initial β that randomizes spins.

Please also see the last paragraph of Section III where we clarified the initial condition dependence between shallow and deep PAOA circuits.

(2) In the present manuscript, it is not clear to what extent the advantages of PAOA are real or are they the artifacts of carefully chosen initial conditions.

AUTHOR RESPONSE

In our work, we presented two results where PAOA demonstrates advantages over existing methods and neither shows any dependence on initial conditions. They are robust and real.

The first result is our comparison against QAOA in Fig. 4, which was explicitly designed to test for generalization. The protocol involved optimizing the PAOA schedules on a dedicated training set of instances. We then applied the learned average schedule to a **fully disjoint set of 30 random test instances without any retraining or fine-tuning**. The superior approximation ratio achieved by PAOA across all depths (Fig. 4b) is therefore a demonstrated advantage on **unseen** problem instances, not an artifact of carefully chosen conditions.

The second result is our analysis of the SK–Lévy model in Fig. 5, **now completely revised**, which shows that these advantages hold in deep circuits where initial condition dependence is not a factor:

- 1. Irrelevance of initial conditions:** For the deep circuits used in this test, both the SA and PAOA schedules begin at very low inverse temperatures ($\beta \approx 0$), corresponding to an initial high-temperature phase. This standard annealing practice ensures the system is thoroughly randomized, erasing any memory of the specific starting spin configuration and making the subsequent performance independent of the initial state.
- 2. Generalization across scale:** The advantage of the multi-schedule heuristic discovered by PAOA is not tied to a specific problem instance or size. The strategy of using distinct schedules for “heavy” and “light” nodes consistently outperforms the best single-schedule SA. This demonstrates that the advantage is a robust feature of the learned strategy itself.

In both cases, the advantages of PAOA are shown to be genuine and generalizable, not artifacts of the initial conditions. Please see Author Action below to see the concrete changes we made to the manuscript to better clarify these points.

AUTHOR ACTION

- Please see the second to last paragraph of Section IIC

- Please see our completely revised Fig. 5, Section VII, as well as Fig. 4.

(3) I find the QAOA comparison misleading. PAOA has already been compared to QAOA by Weitz et al, so why the authors repeat similar discussion here?

AUTHOR RESPONSE

Our comparison of PAOA to QAOA is unique and is not a repeat of prior work. The comparison by Weitz et al. focused on the MaxCut problem for specific graph types. To the best of our knowledge, our work represents the **first** direct comparison of PAOA and QAOA on the **Sherrington-Kirkpatrick (SK) model**, which is a canonical, mean-field spin glass problem and a celebrated benchmark in numerous foundational QAOA studies. A rigorous performance comparison on this specific problem is therefore a critical and previously unaddressed question.

Furthermore, our work is not a repetition but a substantial generalization that validates and extends the PAOA framework to new problem domains and into a scalable, hardware-realized algorithm, as the original authors of Ref.[1], Weitz et al., explicitly acknowledge in their own publication when citing our work (boldface ours):

*“Second, Abdelrahman et al. [44] **extend our theoretical framework and realize the PAOA experimentally**. They developed hardware capable of executing the PAOA, including the classical optimizer, on graphs with up to 500 vertices. This provides independent experimental validation of the PAOA and **a substantial generalization of our theoretical results.**”*

(4) Did the authors find an example that shows QAOA superiority? It would be good to see examples that point towards QAOA advantage as well.

AUTHOR RESPONSE/ACTION

While our objective is to conduct a rigorous, iso-parametric benchmark of PAOA on a canonical (and not previously studied) QAOA benchmark, we agree that a fair assessment with respect to QAOA is absolutely essential.

We now expand the last paragraph of Section VI in the revised manuscript to better illustrate the fundamental possible advantages of QAOA where we state that

“... that a clear quantum advantage over classical algorithms is expected to emerge in problems where interference plays an explicit role, such as in the synthetic examples constructed by Montanaro et al [12]. In these cases, the superposition of many computational paths, enabled by either the problem structure or the quantum algorithm, creates interference patterns that amplify the correct solution while suppressing others, an effect that classical probabilistic samplers struggle to reproduce due to the well-known “sign problem” [32]. As shown in [33], a probabilistic formulation of generic unitary evolutions yields complex (often purely imaginary) weights, so sampling effectively becomes randomized. While post-phase corrections recover unbiased estimates, the number of samples grows exponentially. These interference-dominated regimes are precisely where we do not expect PAOA, or any classical sampler, to compete. On the other hand, unless such interference-driven advantages are clearly identified and demonstrated more broadly across practical problem classes, we do not expect any significant quantum advantage from QAOA. Meanwhile, PAOA offers a rigorous, scalable and high-performing benchmark for variational sampling at problem sizes beyond the reach of current quantum devices.

(5) The message coming from the PAOA-vs-QAOA section seems to be “PAOA is a practical baseline” – this has been established before.

AUTHOR RESPONSE

As we discussed in our response to (3), PAOA vs QAOA on the canonical SK problem has **not** been established before. Moreover, the conclusion the reviewer is drawing, namely, PAOA is a practical baseline is incorrect, as we discuss in our response to (4): PAOA is a superior baseline (not just a practical one) for the SK problem.

(6) I find parts of the presentation ambiguous. The authors repeatedly claim that simulated annealing emerges as a limiting case of their approach. Is PAOA then just a generalization of simulated annealing or is there a new class of problems that PAOA can solve better?

AUTHOR RESPONSE

Simulated Annealing is indeed a sub-case of the general PAOA framework. However, PAOA is not *just* a generalization of SA; it is a more powerful framework that can solve certain classes of problems better by learning heuristics that are inaccessible to standard SA. Our new results in FIG. 5 provide a clear demonstration of this advantage. To summarize:

- SA is a specific, constrained sub-case within the broader PAOA framework. When PAOA is restricted to a single global schedule and run for a sufficient number of layers, it learns to recover the behavior of SA.
- The true advantage of PAOA emerges in problems with underlying structure, such as the Sherrington-Kirkpatrick (SK) model with heavy-tailed Lévy bonds. Standard SA is limited to applying a single, global annealing schedule, which is a suboptimal strategy for such a heterogeneous problem. In contrast, PAOA can utilize a **double-schedule ansatz**. As shown in our new FIG. 5(b), PAOA automatically discovers a problem-aware heuristic by learning to assign a slower, higher-temperature schedule to the “heavy” nodes and a faster-cooling schedule to the “light” nodes. This learned strategy results in a clear and statistically significant performance advantage. As shown in FIG. 5(d), the double-schedule PAOA achieves a consistently higher success probability than the best-performing single-schedule SA, outperforming it on a clear majority of the 500 tested instances.

We believe the double-schedule example we show is the tip of the iceberg: one can imagine many other parameterizations of PAOA, such as private temperatures to each node, improving replica-based algorithms such as simulated quantum annealing and parallel tempering, or entirely new heuristics. The main contribution of our work is to show PAOA can provide a systematic method of exploring these methods by approximating *derivatives* of very high dimensional Markov matrices. We hope that our foundational work will motivate such future studies.

AUTHOR ACTION

Please see our revised Fig. 5 and Section VII.

(7) I am trying to understand whether this approach is fundamentally different or an incremental improvement over old techniques. The paper fails to convincingly establish the approach as a genuinely new paradigm. The presentation and results suggest that the method is an optimized heuristic that may work on carefully chosen benchmarks and only when properly initialized.

AUTHOR RESPONSE/ACTION

As discussed earlier in several of our responses (e.g., (2)-(3)), PAOA does not exhibit initial condition dependence in general.

Moreover, PAOA is an entirely new framework that generalizes Simulated Annealing enabling countless other parameterizations in a principled *optimization* framework.

Our core contributions over Ref. 1 are:

- A global Markov-flow formulation and variational principle linking derivative-free updates to the gradient of the full (exponentially large) Markov process.
- A scalable hardware realization (FPGA) with on-chip annealing and massively parallel sampling.
- New iso-parametric SK benchmarks versus QAOA with generalization across instances and sizes.
- Automatic discovery of structured, heterogeneous schedules in heavy-tailed SK–Lévy that outperform single-schedule SA.

We genuinely believe PAOA is a fertile ground. QAOA has a vast literature with thousands of papers; by contrast, PAOA is discussed in just a single one so far (excluding ours). We believe PAOA can be a serious contender and merits broader exploration.

One final note: We were alerted by the Editor that our current title needs to be modified, as such we are proposing “Generalized Probabilistic Approximate Optimization Algorithm” as a new title of the paper.

II. REVIEWER #2

The paper introduces the Probabilistic Approximate Optimization Algorithm (PAOA), a new variational Monte Carlo framework. PAOA was inspired by the Quantum Approximate Optimization Algorithm (QAOA) and thus is a “dequantized” version of QAOA. The authors show that PAOA generalizes simulated annealing and can rediscover annealing schedules. They demonstrate hardware acceleration on an FPGA p-bit computer with large performance gains over CPUs. PAOA is benchmarked against QAOA on the Sherrington–Kirkpatrick model and achieves higher approximation ratios under matched parameters. The work is significant because it provides a scalable classical baseline and a tool for discovering new non-equilibrium heuristics, with impact on optimization and probabilistic computing.

AUTHOR RESPONSE

We sincerely thank the reviewer for their thoughtful review and positive assessment. We believe we have addressed each of the points in the revised manuscript, and a color-tracked `diff` file highlighting every change is appended for the reviewer’s convenience.

(1) Definition of Symbols and Notation. Define N in the abstract and again in Section II. From my understanding N = number of spins/variables in the optimization problem, and 2^N = size of the full state space.

AUTHOR RESPONSE/ACTION

The reviewer is correct about the definition. We removed the explicit mention of N from the abstract for clarity and added descriptions in the caption of Fig. 1 in Section II.

(2) In Fig. 1, explicitly define symbols: ρ , $\hat{\rho}$, p , M , m . Clarify upfront why M (ansatz size, possibly with hidden variables) differs from N (problem size). This was explained later in Section II.C, but until I read that part I was confused.

AUTHOR RESPONSE/ACTION

We agree. We have revised the Fig. 1 caption to explicitly define the symbols ρ , $\hat{\rho}$, p , M , and m , and clarified why the ansatz size M can differ from the problem size N . We have also added an explicit discussion at the beginning of Section II so that this distinction is clear upfront rather than deferred to Section II.C.

(3) In Section IV, is “best beta” the optimization criterion defined as minimization of expected energy, i.e., $\text{best} \equiv \min\langle E \rangle$?

AUTHOR RESPONSE

We agree. In Sec. IV the optimizer updates the schedules by minimizing the *average sampled energy* $\langle E \rangle$ over N_E independent runs. After training, each candidate schedule is evaluated on a separate batch of 10^5 inference runs. “Best β ” denotes the schedule whose inference runs yield the highest success probability, where success probability is defined as the fraction of runs that reach the putative ground-state energy at the final layer. The success probabilities shown in the insets of Fig. 3(c) are computed independently for each optimizer run under its corresponding schedule.

AUTHOR ACTION

We have written the third paragraph of Section IV to better clarify these points.

(4) Numerical anomalies? In Fig. 3, the higher energy observed at $p = 1$ for 15 layers is unexplained. Is this just a numerical artifact? Some comment would help the reader.

AUTHOR RESPONSE

The reviewer is likely referring to the inverse temperature schedule, $\beta(p)$, rather than to the energy itself. One might reasonably expect an optimal β schedule must be monotonically increasing over the layer depth, p . In practice, however, we observe that there is a large *degeneracy* of β schedules. The shaded blue lines in Fig. 3 show all the schedules found by the optimizer and the insets report the corresponding success probabilities. The near-uniform success probabilities indicate that there are many degenerate and near-equivalent schedules. Despite the apparent outlier at its second point, the red schedule at $p = 15$ provides the highest success probability among 100 independent optimizer runs.

AUTHOR ACTION

We have revised the discussion around Fig. 3 to include this observation.

(5) In Fig. 3(b), I noticed an apparent majorization ordering across circuit depths. Ordering probabilities from largest to smallest gives $P_{15} \succ P_{10} \succ P_5$, suggesting structured improvement beyond average energy. Since entropy and purity are Schur-convex/concave, one expects $H(P_{15}) < H(P_{10}) < H(P_5)$. Entropy and purity from truncated data may not show this, but the full distributions likely would. This indicates that adding layers reduces entropy and concentrates probability on the correct answer. This may or may not be an interesting additional insight for the manuscript.

AUTHOR RESPONSE

The reviewer makes a deep and insightful observation. We agree that the full distributions (that are out of reach) likely would show explicit majorization. It may also be possible to make mathematical statements about this majorizing behavior in the appropriate limit. We hope that the small number of PAOA papers (starting with the one our work built on Ref. [1]) will motivate such studies further.

AUTHOR ACTION

We added the following paragraph to Section IV:

As depth p increases, our schedules both cool further and spend more total sweeps at lower temperatures, so the energy histograms naturally put more weight near the ground state. In majorization language, the deeper runs appear to majorize the shallower ones, but we only show energy marginals, therefore, this is suggestive rather than proof. Similar cooling-with-depth trends were observed for QAOA Ref[27].

(6) The y-axis in Fig. 4. The scale seems a little off. Is it logarithmic, or is there a label typo from Illustrator editing? The spacing between $|-0.8 - (-0.6)| \neq |-0.6 - (-0.4)|$. Regardless, I wonder if clipping the y-axis to -0.3 might allow for the data to look less compressed. Or rescaling (possibly log scale) to improve readability. [This also applies to Figure S5]

AUTHOR RESPONSE/ACTION

We agree. We revised the main and supplementary figures and used linear scaling for clarity. Following the excellent suggestion by the reviewer, we also clipped y-axis appropriately for the main and supplementary figures.

(7) Fig. 5e. I'm trying to get a feel for the improvement of PAOA over SA. Are the differences shown meaningful? Or am I missing the point of what is illustrated by the plots here?

AUTHOR RESPONSE

In light of this comment, we have **completely revised** Fig. 5 and the associated Section VII. Even though the improvement of double-schedule PAOA over standard (single schedule) SA is statistically significant in our original

Fig. 5, results did not show an appreciable difference between the two algorithms. We suspect the reason is the *shallow* depth (p) we used: the differences between the anneals did not have enough “time” to grow.

To better show whether the double-schedule provided by PAOA improves SA at deeper layers, we extrapolate the schedules provided by the optimizer, using a straightforward procedure. With more depth, we observe a clearer increase in success probability (from about 16% to 20%) for the double-schedule PAOA over the single SA, for this class of heavy-tailed SK problem.

AUTHOR ACTION

Please observe the new Fig. 5 in the marked up manuscript along with the discussion.

(8) Reference [31] should be corrected/clarified: Make it clear that it is Chapter 10 in Simulated Annealing, Emile Aarts, Jan Korst, and Wil Michiels.

AUTHOR RESPONSE/ACTION

We thank the reviewer for catching a serious citational error: our original reference was to *another* book by Emile Aarts and Jan Korst. We now corrected the citation in the revised manuscript.

One final note: We were alerted by the Editor that our current title needs to be modified, as such we are proposing “Generalized Probabilistic Approximate Optimization Algorithm” as a new title of the paper.

Generalized Probabilistic Approximate Optimization : A New Variational Monte Carlo Algorithm

Abdelrahman S. Abdelrahman,¹ Shuvro Chowdhury,¹ Flaviano Morone,² and Kerem Y. Camsari¹

¹*Department of Electrical and Computer Engineering,
University of California, Santa Barbara, Santa Barbara, CA, 93106, USA*

²*Center for Quantum Phenomena, Department of Physics,
New York University, New York, New York 10003 USA*

We introduce a generalized *Probabilistic Approximate Optimization Algorithm (PAOA)*, a classical variational Monte Carlo framework that extends and formalizes prior work by Weitz *et al.* [1], enabling parameterized and fast sampling on present-day Ising machines and probabilistic computers. PAOA operates by iteratively modifying the couplings of a network of binary stochastic units, guided by cost evaluations from independent samples. We establish a direct correspondence between derivative-free updates and the gradient of the full $2^N \times 2^N$ Markov flow Markov flow over the exponentially large state space, showing that PAOA admits a principled variational formulation. Simulated annealing emerges as a limiting case under constrained parameterizations, and we implement this regime on an FPGA-based probabilistic computer with on-chip annealing to solve large 3D spin-glass problems. Benchmarking PAOA against QAOA on the canonical 26-spin Sherrington–Kirkpatrick model with matched parameters reveals superior performance for PAOA. We show that PAOA naturally extends simulated annealing by optimizing multiple temperature profiles, leading to improved performance over SA on heavy-tailed problems such as SK–Lévy.

I. INTRODUCTION

Monte Carlo algorithms remain a central tool for exploring complex energy landscapes, especially in the context of combinatorial optimization and statistical physics. Classical methods such as simulated annealing (SA) have been widely applied across these domains, but their reliance on slowly equilibrating processes limits their performance on rugged energy landscapes [2–7]. New approaches are needed to construct non-equilibrium strategies that retain algorithmic simplicity while improving solution quality.

Inspired by the quantum approximate optimization algorithm (QAOA) [8–9, 11–18] [8–18], Weitz *et al.* [1] proposed a classical variational protocol based on the direct parameterization of low-dimensional Markov transition matrices. Their work introduced the term Probabilistic Approximate Optimization Algorithm (PAOA), raising the possibility of classical, variational analogs to QAOA within probabilistic architectures.

Building on this foundational work, we formalize and generalize PAOA. We move beyond the original proposal’s edge-local matrices to derive a global $2^N \times 2^N$ Markov-flow formulation applicable to any k -local Ising Hamiltonian of size N . This framework unifies a wide spectrum of variational ansätze, from global and local schedules to fully-parameterized couplings, and allows them to be stacked to an arbitrary depth p . Crucially, we connect this variational theory to practice by building on the p-computing framework [19], where networks of binary stochastic units (p-bits) sample from Boltzmann-like distributions through asynchronous dynamics. This model has a demonstrated record of success in optimization and inference tasks [20–21, 23] [20–23] and provides a natural substrate for our work. By implementing these dynamics on an FPGA, we demonstrate for the first time a scalable, hardware-based path for executing the PAOA.

We show that PAOA admits a broad class of parameterizations, including global, local, and edge-specific annealing schedules. Within this framework, SA emerges as

a limiting case under constrained schedules. We implement this regime on an FPGA-based p-computer to demonstrate large-scale, high-throughput sampling. In contrast to standard SA, PAOA’s flexible parameterization enables the discovery of non-equilibrium heuristics that exploit structural features of the problem. One such example is demonstrated in heavy-tailed Sherrington–Kirkpatrick (SK) models, where PAOA assigns higher effective temperatures to strongly coupled nodes and lower temperatures to weakly connected ones, enabling annealing profiles with better performance over vanilla SA. This adaptive scheduling capability provides a powerful framework for the automated discovery of novel annealing heuristics.

We also benchmark PAOA against QAOA on the SK model using matched parameter counts and observe superior performance. This may not be surprising, as the practical performance advantages of QAOA over classical and probabilistic alternatives remain uncertain. A clear quantum advantage is expected to emerge in problems where interference plays an explicit role, such as in the synthetic examples constructed by Montanaro *et al.* [13]. Our work builds on the promising results of Weitz *et al.* [1] on the max-cut problem by expanding PAOA into a scalable, hardware-compatible, and general-purpose optimization algorithm.

The remainder of this paper is organized as follows. In Sec. II, we present the theoretical formulation of PAOA, including its parameterization, Markov structure, and sampling-based approximation with p-bits. In Sec. III, we study a simple majority gate problem to illustrate the correspondence between analytical gradients and derivative-free optimization. In Sec. IV, we demonstrate the recovery of SA as a special case and implement this regime on hardware. In Sec. V, we describe the FPGA architecture used to accelerate sampling. In Sec. VI, we benchmark PAOA against QAOA on the SK model using matched parameter counts. In Sec. VII, we study a heavy-tailed SK variant to show how PAOA can discover multiple annealing schedules that outperform standard single-schedule simulated annealing.

FIG. 1. Overview of PAOA. A hybrid classical-probabilistic-classical-probabilistic architecture iteratively updates a weight matrix J and bias vector h using feedback from a probabilistic computer. The p -computer samples from a distribution defined by (J, h) , approximating the exact Markov flow $\rho_{p+1} = W \rho_p$, where $W \in \mathbb{R}^{2^M \times 2^M}$. The resulting samples drawn from $\hat{\rho}_p$ are used to evaluate a cost function, which the classical optimizer minimizes by adjusting the variational parameters. Here, ρ denotes the exact probability distribution over spin configurations, $\hat{\rho}$ the sampled approximation, p the layer number, M the number of total spins represented in the ansatz (including possible hidden variables), and m a specific spin configuration (state). The cost function is typically the energy of a spin-glass spin glass mapped to an optimization problem (e.g., Eq. 3) but it can also be a likelihood function if PAOA is learning from data (e.g., Eq. 5). Importantly, the ansatz is independent of size M can exceed the original problem size and can support M spins ($M \geq N$ in general, with $M = N$ since hidden variables used in may be introduced to increase the ansatz) and graph topologies representation. In this work, we use $M = N$ in all experiments unless noted. At convergence ($p = k$), the distribution concentrates around low-energy solutions.

II. THEORY OF PAOA

The power of the Probabilistic Approximate Optimization Algorithm (PAOA) comes from a conceptual departure from traditional optimization methods like simulated annealing (SA). While SA explores a single, fixed energy landscape defined by a model Hamiltonian, PAOA treats the landscape itself as a variational object to be optimized. The goal is to dynamically reshape this landscape to make the ground state of the original problem easier to find.

This creates a two-level optimization loop, as illustrated in FIG. 1. In an *outer loop*, a classical optimizer adjusts a set of variational parameters, θ . In an *inner loop*, a probabilistic computer samples states from the variational landscape currently parametrized by θ . These samples are then used to evaluate the energy function of the original problem, providing feedback to the classical optimizer, which updates θ by optimizing an auxiliary θ . In the most general case, θ specifies the full set of couplings and fields $\{J^{(k)}, h^{(k)}\}$ at each layer k , so that the landscape itself is directly reconfigured by the optimizer. Simpler ansätze, such as global or node-wise schedules introduced in Sec. III A, can be viewed as constrained subsets of this general J, h

parameterization (for example, a global inverse temperature amounts to scaling all entries of J by a single parameter). The sampler acts on M spins in the ansatz, while the original problem has N visible spins. In general $M \geq N$ because hidden spins may be introduced to enrich the representation, in direct analogy to neural quantum states (NQS) [24]. Unless otherwise stated, we set $M = N$ in the remainder of the paper, for simplicity, and leave the possibility of introducing hidden spins for better representation to future work.

During training, samples generated under θ are evaluated with a task-dependent cost function $\mathcal{L}(\theta)$ to begin the next iteration. This framework decouples the problem's energy function. In problems with a known target set of states (e.g., majority logic gate, discussed in Section III), \mathcal{L} is typically chosen as a negative log-likelihood. In optimization problems such as 3D spin glasses, \mathcal{L} is instead taken as the average energy over sampled configurations. In either case, feedback from the samples updates θ , decoupling the problem energy from the sampling dynamics, allowing the algorithm to learn efficient, and enabling PAOA to discover non-equilibrium pathways to a solutions.

A. A Spectrum of Variational Ansätze

The power and flexibility of PAOA lie in the choice of the variational parameters $\underline{\theta}$ that define the search landscape. We refer to this parameterization as the variational ansatz. Let \$N\$ denote the total number of spins in the problem. To explore the tradeoffs between expressiveness and generalization, in this paper we study four representative ansätze, defined by the number of free parameters, Γ :

$$\Gamma \in \left\{ 1, 2, N, \frac{N(N-1)}{2} \right\}$$

- When $\Gamma = 1$, the landscape is scaled by a single, learned inverse temperature $\beta^{(k)}$ at each layer k . This is the global annealing schedule, which is the closest analog to SA, with the crucial difference that the schedule is learned rather than predefined.
- When $\Gamma = 2$, two independent schedules are assigned to distinct parts of the graph, typically divided based on node properties like weighted degree.
- When $\Gamma = N$, each node is assigned a local schedule $\beta_i^{(k)}$, allowing the algorithm to anneal different parts of the problem at different rates.
- When $\Gamma = N(N-1)/2$, the entire symmetric coupling matrix $J_{ij}^{(k)}$ is used as the variational ansatz, giving the algorithm maximum freedom to reshape the landscape at each layer.

Recent work in QAOA has shown that increasing the number of variational parameters per layer can improve convergence and reduce the required circuit depth [9][10]. A similar principle applies to PAOA: more expressive schedules allow the system to reach useful non-equilibrium distributions in fewer layers, but may also increase the risk of overfitting. The balance between expressiveness and generalization remains a key consideration when selecting an ansatz.

B. Implementation via Probabilistic Computers

The practical implementation of the sampling inner loop is achieved using a probabilistic computing (p-computing) framework [19]. This framework uses networks of binary stochastic units, or p-bits, that sample from Boltzmann-like distributions via asynchronous updates governed by Glauber dynamics. A single p-bit updates its state m_i according to

$$m_i = \text{sgn}[\tanh(\beta I_i) - \text{rand}_u(-1, 1)] \quad (1)$$

where $\text{rand}_u(-1, 1)$ is a uniform random variable, β is the inverse temperature from the variational ansatz, and the local input I_i is

$$I_i = \sum_j J_{ij} m_j + h_i \quad (2)$$

As long as connected p-bits are updated sequentially, in any random order, this update rule generates samples from a

Boltzmann distribution $P(\{m\}) \propto \exp[-\beta E(\{m\})]$ over time, associated with the energy

$$E(\{m\}) = - \sum_{i < j} J_{ij} m_i m_j - \sum_i h_i m_i \quad (3)$$

C. Formalism and Connection to Markov Chains

While in practice PAOA relies on direct MCMC sampling, the process has a rigorous foundation in the theory of Markov chains. The evolution of the probability distribution ρ over the state space can be described by a sequence of transition matrices. To formalize this, we consider a p -layer process where the distribution after k layers, ρ_k , is given by

$$\rho_k = W^{(k)}(\underline{\theta}^{(k)}) \dots W^{(1)}(\underline{\theta}^{(1)}) \rho_0 \quad (4)$$

where $\rho_0 \in \mathbb{R}^{2^N}$ is the initial distribution and each $W^{(k)}$ is a $2^N \times 2^N$ transition matrix parameterized by the variational parameters $\underline{\theta}^{(k)}$ for that layer. For a sequential update scheme, each $W^{(k)}$ can be factorized as a product of single-site update matrices, $W^{(k)} = w_N^{(k)} w_{N-1}^{(k)} \dots w_1^{(k)}$. The exact construction of these matrices from the p-bit update rule is detailed in the Supplementary Information.

This layered application of stochastic matrices to a probability vector ρ is the direct classical analog of applying unitary operators $U^{(k)}$ to a wavefunction ψ in QAOA. The crucial distinction, however, is that each $W^{(k)}$ is a norm-one-preserving stochastic matrix that mixes non-negative probabilities, whereas each $U^{(k)}$ is a norm-two-preserving unitary matrix that rotates complex vectors.

The absence of complex phases in the classical evolution means that PAOA proceeds without the possibility of quantum interference. Consequently, for optimization problems where a constructive interference mechanism for QAOA is unclear, its advantage over classical alternatives is not guaranteed. Indeed, as we will demonstrate in Sec. VI, our benchmarking shows that PAOA consistently reaches higher-approximation ratios on the Sherrington-Kirkpatrick model when compared to QAOA using an identical number of variational parameters.

The ultimate objective of PAOA is to find the states $\{m\}$ that minimize the problem's energy function (Eq. 3). This is achieved through the two-level optimization loop described earlier. The outer loop minimizes a cost function (different from the problem's energy function) over the variational parameters $\underline{\theta}$, thereby reshaping the sampling landscape to make the optimal states $\{m\}$ more probable and easier to find through MCMC sampling in the inner loop.

For a target set of states \mathcal{X} , this cost function can be the negative log-likelihood:

$$\mathcal{L}(\underline{\theta}) = - \sum_{\{m\} \in \mathcal{X}} \ln(\rho_p(\{m\}; \underline{\theta})) \quad (5)$$

In practice, as the exact computation of ρ_p is intractable for large N , it is replaced by an empirical distribution $\hat{\rho}_p$ estimated from N_E independent MCMC samples. Because the samples are generated from a shallow Markov process (small p), the system typically remains out of equilibrium.

FIG. 2. Majority gate benchmark: comparing analytical and sampling-based PAOA. (a) Fully connected four-node network used to implement the majority gate $Y = \text{MAJ}(A, B, C)$, where $Y = A \vee B$ if $C = 1$ and $Y = A \wedge B$ if $C = 0$. The table lists the eight valid input-output combinations, labeled by their decimal encoding. (b) Training loss during optimization, comparing exact gradients (blue) to gradient-free optimization using COBYLA (orange) with 10^7 MCMC samples. Both methods converge to the same minimum. (c) Time evolution of the exact distribution over two layers using analytical Markov matrices. Initial uniform distribution ($p=0$) is transformed into a peaked distribution ($p=2$) concentrated on the correct truth table entries. (d) Corresponding evolution using MCMC samples and COBYLA. The approximated distributions closely match the exact dynamics.

This non-equilibrium character distinguishes PAOA from classical annealing methods and may unlock new features such as initial condition dependence and faster convergence to the solution. However, this may also become a liability, for shallow circuits, as strong dependence on the initial distribution ρ_0 or overfitting may prevent the learned heuristics from generalizing. Therefore, the same ρ_0 used during training should also be used during inference.

Deep PAOA circuits do not exhibit any initial condition dependence especially with small initial β that randomizes spins (Section IV).

Finally, the PAOA framework is general in two key respects. First, the cost function being minimized is not restricted to the two-local form, such as the Ising energy of Eq. (3); it can be any computable function over the states $\{m\}$, including higher-order k -local Hamiltonians or likelihood

functions for machine learning tasks. Second, the variational landscape used for sampling, while implemented here with an Ising-like p-computer, is also not fundamentally limited. The PAOA approach is compatible with any model from which one can efficiently draw samples via MCMC, such as Potts models.

~~**Majority gate benchmark: comparing analytical and sampling-based PAOA. (a) Fully connected four-node network used to implement the majority gate $Y = \text{MAJ}(A, B, C)$, where $Y = A \vee B$ if $C = 1$ and $Y = A \wedge B$ if $C = 0$. The table lists the eight valid input-output combinations, labeled by their decimal encoding. (b) Training loss during optimization, comparing exact gradients (blue) to gradient-free optimization using COBYLA (orange) with 10^7 MCMC samples. Both methods converge to the same minimum. (c) Time evolution of the exact distribution over two layers using analytical Markov matrices. Initial uniform distribution ($p=0$) is transformed into a peaked distribution ($p=2$) concentrated on the correct truth table entries. (d) Corresponding evolution using MCMC samples and COBYLA. The approximated distributions closely match the exact dynamics.**~~

III. REPRESENTATIVE EXAMPLE: MAJORITY GATE

To illustrate the behavior of PAOA under both exact and approximate dynamics, we solve a small optimization problem involving a four-node majority gate. This problem serves as a tractable testbed for understanding the role of the classical optimizer.

The majority gate is defined over four binary variables $[A, B, C, Y]$, where the output Y is given by

$$Y = \text{MAJ}(A, B, C) = \begin{cases} A \vee B, & \text{if } C = 1 \\ A \wedge B, & \text{if } C = 0 \end{cases} \quad (6)$$

The eight correct input-output combinations, shown in the truth table in FIG. 2a, define the target set of states, \mathcal{X} . The goal of the optimization is to find variational parameters that cause the final distribution, ρ_p , to be concentrated on this set.

As a starting graph, we consider a fully connected Ising graph with $J_{ij} = +1$ and no biases. In this example, the variational parameters are node-specific schedules $\beta_i(p)$, corresponding to $\Gamma = N$. To validate the role of the numerical optimizer, we compare two optimization strategies: a full Markov chain with gradient-based optimization and an MCMC-based approximation using the COBYLA algorithm [25].

For the gradient-based approach, we use the formulation in Sec. III and optimize over two layers ($p=2$) with a uniform initial distribution $\rho_0 = 1/16$. The cost is the negative log-likelihood over the eight correct states, and the variational parameters $\beta_i(p)$ are updated using gradient descent with a fixed learning rate $\eta = 0.004$. Optimization stops when either the gradient norm falls below 10^{-7} or a fixed iteration budget is reached.

In the MCMC-based approach, the same initial condition is used, but sampling is performed using 10^7 independent

runs. COBYLA is used to update the parameters based on the empirical distribution $\hat{\rho}$. The optimization stops based on parameter convergence or a maximum number of function evaluations. Results are shown in FIG. 2b, where the two curves converge to nearly identical minima.

FIG. 2c and d show the final distributions under both methods. The agreement between the exact Markov flow and the sampled histogram confirms that derivative-free optimization closely tracks the true gradient.

Algorithm 1: PAOA: global annealing schedule

Input : number of nodes N , number of layers p , number of sweeps per layer s/p , number of experiments N_E , problem weight matrix J , initial variational parameters β , variational parameters tolerance $\varepsilon_{\text{step}}$, maximum iterations t_{max}

Output: trained annealing schedule β_{opt}

- 1 **Function** p-computer (FPGA) ($\beta, J, N, p, s/p, N_E$) :
- 2 **for** $i \leftarrow 1$ **to** N_E **do**
- 3 initialize all spins randomly
- 4 **for** $j \leftarrow 1$ **to** p **do**
- 5 $\beta \leftarrow \beta(i)$
- 6 **for** $k \leftarrow 1$ **to** s/p **do**
- 7 **for** $l \leftarrow 1$ **to** N **do**
- 8 | solve Eqs. (1) and (2)
- 9 | store p-bit states in the BRAM
- 10 **return** all stored p-bit states
- 11 **Function** PAOA-circuit ($\beta, J, N, p, s/p, N_E$) :
- 12 states \leftarrow p-computer (FPGA) ($\beta, J, N, p, s/p, N_E$)
- 13 compute the energy using Eq. (3) for all states
- 14 compute the average energy
- 15 **return** average energy
- 16 **while** ($\text{step size} > \varepsilon_{\text{step}}$ and $\text{number of iterations} < t_{\text{max}}$) **do**
- 17 average energy \leftarrow PAOA-circuit ($\beta, J, N, p, s/p, N_E$)
- 18 minimize average energy and get a perturbation vector (\mathbf{p}) using a gradient-free optimizer
- 19 **for** $i \leftarrow 1$ **to** p **do**
- 20 $\beta(i)^{t+1} \leftarrow \beta(i)^t + \mathbf{p}(i)$
- 21 $t \leftarrow t + 1$
- 22 **return** optimal variational parameters

We also test the same procedure on a 5-node full-adder circuit using the fully-parameterized $J_{ij}(p)$ ansatz. The resulting distributions (see FIG. S1) once again demonstrate tight correspondence between exact and approximate optimization at each layer.

Although analytical optimization is feasible for small N , the $2^N \times 2^N$ transition matrix becomes intractable for larger systems. The success of COBYLA in this setting confirms that PAOA can be implemented efficiently on general p-computers using standard MCMC.

Interestingly, PAOA can explore parameter regimes not accessible to traditional Ising formulations. For example, optimal β values may be negative, corresponding to negative temperatures without a clear interpretation in the statistical physics-based context. Since equilibrium Boltzmann-sampling is not necessarily the starting point, alternative and

FIG. 3. Discovering simulated annealing with PAOA on a 3D spin-glass problem. (a) The hybrid architecture combines an FPGA-based p -computer for MCMC sampling with a classical CPU that optimizes the global annealing schedule, $\beta^{(k)}$. (b) Energy histograms on a single $L^3 = 6^3$ instance before (red) and after (blue) optimization. The optimized schedule shifts the distribution toward the putative ground state ($E \approx -360$), increasing its discovery frequency. (c) Optimized schedules for p -layer architectures where $p \in \{5, 10, 15\}$. Starting from a flat initial schedule ($\beta = 2$, green), PAOA consistently discovers cooling schedules (best shown in red) that resemble SA. Faint curves show all 100 optimization runs. The insets display the sorted success probabilities, demonstrating that deeper architectures improve the average success probability and reduce run-to-run variability.

more hardware-friendly activation functions could also be explored with PAOA, such as replacing the hyperbolic tangent in Eq. (II) with the error function erf, or with saturating linear functions.

PAOA: global annealing schedule

Discovering simulated annealing with PAOA on a 3D spin-glass problem. (a) The hybrid architecture combines an FPGA-based p -computer for MCMC sampling with a classical CPU that optimizes the global annealing schedule, $\beta^{(k)}$. (b) Energy histograms on a single $L^3 = 6^3$ instance before (red) and after (blue) optimization. The optimized schedule shifts the distribution toward the putative ground state ($E \approx -360$), increasing its discovery frequency. (c) Optimized schedules for p -layer architectures where $p \in \{5, 10, 15\}$. Starting from a flat initial schedule ($\beta = 2$, green), PAOA consistently discovers cooling schedules (best shown in red) that resemble SA. Faint curves show all 100 optimization runs. The insets display the sorted success probabilities, demonstrating that deeper architectures improve the average success probability and reduce run-to-run variability. The shallow nature of the Markov chain used in

PAOA introduces a dependency on the initial distribution ρ_0 . In contrast to standard Boltzmann training methods such as Contrastive Divergence [26], which seek to approximate the equilibrium distribution through extensive sampling, PAOA optimizes the evolution of the distribution under a fixed initialization. This shortcut avoids the computational burden of long mixing times but restricts the learned dynamics to a specific initialization at shallow depth. While this dependence can hinder generalization at shallow depth, as we demonstrate in Section IV for deep PAOA circuits, initialization dependence is not an important concern. Whether this initialization dependence is a liability or a feature depends on context: it can enable targeted sampling for certain inference tasks, but it may also hinder generalization if many different instances need to be optimized at the same time.

IV. DISCOVERING SIMULATED ANNEALING WITH PAOA

An intriguing question is whether PAOA, when restricted to a global annealing schedule with one parameter per time

step, can recover the well-known structure of simulated annealing (SA). In this section, we show that the answer is yes. Using a minimal parameterization with a single global inverse temperature, $\beta^{(k)}$, for each one of the p total layers, PAOA is able to discover SA-like profiles that optimize the average energy on a large three-dimensional (3D) spin-glass instance.

TABLE I. Parameters used in Algorithm 1.

Parameter	Value
number of nodes (N)	216
number of Monte Carlo Sweeps (MCS) /layer (s/p)	720
number of layers (p)	{5, 10, 15}
initial variational parameters ($\beta(p)$)	2
variational parameters tolerance (ϵ_{step})	10^{-4}
maximum iterations (t_{max})	5000
number of independent experiments (N_E)	10^5

We apply PAOA to a 3D cubic lattice of size $L^3=6^3$, which yields a sparse bipartite interaction graph. To leverage this structure for massive parallelism on our FPGA-based p-computer, we use chromatic updates [21]–[22]. We employ the simplest $\Gamma=1$ global annealing ansatz, where all nodes share the same scalar inverse temperature $\beta^{(k)}$ at each layer k , forming a schedule of total length $p \in \{5, 10, 15\}$.

Sampling is performed on hardware using extensive-720 Monte Carlo sweeps per layer. The initial distribution is uniform. We choose a uniform initial distribution, achieved by appending $\beta=0$ to the beginning of the schedule. After sampling, configurations are transferred to the CPU where a classical optimizer the CPU evaluates the average energy $\langle E \rangle$ using Eq. (3) and COBYLA updates the schedule $\beta^{(k)}$ using COBYLA. Notice that the role of the $\beta^{(k)}$. In this training loop the cost function is played here by the average sampled energy, $\langle E \rangle$, treated by the classical optimizer as a function of the variational parameters $\beta^{(k)}$, estimated from N_E independent MCMC runs. After training, however, each candidate schedule is evaluated on a fresh batch of 10^5 runs and assigned a success probability, defined as the fraction of runs that reach the putative ground-state energy at the final layer. When referring to the “best” schedule or “best β ” in FIG. 3, we mean the schedule that achieves the highest success probability out of this independent batch. Insets in FIG. 3(c) show these success probabilities for all 100 optimizer runs, illustrating both the variability across runs and the improvement with circuit depth.

The loop continues until either convergence or a maximum number of energy evaluations is reached. The algorithm is outlined in Algorithm 1 and parameters are listed in Table I.

The results of this optimization are summarized in FIG. 3. The optimized schedules shift the energy distribution toward lower values (FIG. 3(b)), increasing the probability of reaching the putative ground state at $E=-360$, which was independently found using a linear SA schedule with one million MCSs (from $\beta=0.1$ to 5) schedule with 10^6

MCS. The schedules responsible for this improvement are shown in FIG. 3(c). Notably, without any constraints on monotonicity, PAOA consistently learns schedules that begin at high temperature (low β) and gradually cool (increase β), resembling classical SA despite starting from a flat $\beta=2$ schedule. The

As depth p increases, our schedules both cool further and spend more total sweeps at lower temperatures, so the energy histograms naturally put more weight near the ground state. In majorization language, the deeper runs appear to majorize the shallower ones, but we only show energy marginals, therefore, this is suggestive rather than proof. Similar cooling-with-depth trends were observed for QAOA [28].

The elevated β at the first step ($p=1$) for the 15-layer case may be a result of the substantial degeneracy among near-optimal schedules. Many distinct β profiles achieve nearly identical success probabilities (shaded blue lines in FIG. 3(c)). The inset histograms show that while individual schedule shapes vary across the schedules vary across 100 optimizer runs, the performance remains stable, especially for (even for non-monotonic schedules) with the smallest variance in success probabilities for deeper schedules ($p=15$: $p=15$).

Constraints could be added to restrict the class of allowed schedules, e.g., enforcing linear, exponential, or monotonic cooling profiles using inequality constraints in COBYLA. We did not use such constraints to avoid biasing the solution toward SA. These results demonstrate that SA emerges as a special case of variational Monte Carlo, recoverable from the global schedule ansatz. Moreover, PAOA proves to be more general, as it frequently finds alternative, points to the possibility of alternative non-monotonic schedules with comparable performance, indicating the flexibility of the algorithm.

PAOA vs QAOA on the Sherrington-Kirkpatrick model. (a) PAOA results using a two-schedule ansatz with $2p$ parameters compared against QAOA. For each depth p , the PAOA schedules are optimized on a separate training set; the average schedule is then applied to 30 random test instances of size $n=26$ without retraining. QAOA results use optimal parameters from prior work [8, 27]. Red crosses denote averages across the 30 instances, blue dots show individual instance energies, and the solid green line indicates the average ground-state energy per spin. (b) Approximation ratios of PAOA and QAOA, averaged across the 30 instances. Error bars indicate the 95% confidence intervals computed from 10^4 bootstrap samples with replacement.

Constraints could be added to restrict the class of allowed schedules, e.g., enforcing linear, exponential, or monotonic cooling profiles using inequality constraints in COBYLA. We did not use such constraints in this example to avoid biasing the solution toward SA.

V. PAOA ON FPGA WITH ON-CHIP ANNEALING

Wall-clock time for 3D spin glass ($L^3=6^3$) with varying layer count. Layers 5–10–15 Flips/ns Time (MCS/replica)

FIG. 4. Parameters used in Algorithm 1. Parameter Value number of nodes (N) 216 number of Monte Carlo Sweeps (MCS) 720 layer (s/p) 5 number of layers (p) 15 PAOA results using a two-schedule ansatz with $2p$ parameters compared against QAOA. For each depth p , 40 the PAOA schedules are optimized on a separate training set; the average schedule is then applied to 30 random test instances of size $N=26$ without retraining. QAOA results use optimal parameters from prior work [8, 27]. Red crosses denote averages across the 30 instances, 15} initial variational parameters ($\beta(p)$) blue dots show individual instance energies, and the solid green line indicates the average ground-state energy per spin. (b) 2 variational parameters tolerance (ϵ_{step}) 10^{-4} maximum iterations (M) 5000 number of independent experiments (N_E) 10^5 Approximation ratios of PAOA and QAOA, averaged across the 30 instances. Error bars indicate the 95% confidence intervals computed from 10^4 bootstrap samples with replacement.

3.6×10^7 7.2×10^7 1.08×10^8 CPU (s) 1953 3800 5850
0.0398 FPGA (s) 2.46 4.90 7.36 31.74

To support long annealing schedules in 3D spin-glass problems, we designed a p -computer architecture that performs annealing entirely on-chip, unlike earlier implementations that require off-chip resources for annealing (e.g., [21]). The annealing schedule $\beta^{(k)}$ is preloaded to the FPGA and indexed via a layer counter that advances after a fixed number of Monte Carlo sweeps (MCS). At each step, the current β value is used to scale the synaptic input I_i via a digital signal processing (DSP) multiplier (see FIG. S4).

The full p -layer annealing process runs *uninterrupted* on the FPGA, and the final spin configurations are read out only after the last layer is complete.

The FPGA operates in fixed-point arithmetic with $s[4][5]$ $s\{4\}\{5\}$ precision (where s denotes the sign bit, followed by 4 integer and 5 fractional bits) for both $\beta^{(k)}$ and J_{ij} . We Due to the size of the FPGA we use, we instantiate 10 independent replicas of the $L^3=6^3$ spin-glass graph, enabling parallel updates of 2160 p -bits per cycle. This on-chip architecture provides a significant performance advantage, achieving an approximately Our FPGA architecture with on-chip annealing capability achieves about approximately an 800-fold reduction in wall-clock time for the entire annealing process in the FPGA compared to an optimized graph-colored

TABLE II. Wall-clock time for 3D spin glass ($L^3=6^3$), comparing CPU to FPGA with 10 identical replicas and 10^4 independent runs.

Layers	5	10	15	Flips/ns
Time (MCS/replica)	3.6×10^7	7.2×10^7	1.08×10^8	
CPU (s)	1953	3800	5850	0.0398
FPGA-10 replicas (s)	2.46	4.90	7.36	31.74

CPU implementation, as detailed in CPU implementation with graph-colored Gibbs sampling (Table II).

PAOA is parallelizable at multiple levels. First, the cost estimator $\langle E \rangle$ is computed from N_E independent MCMC runs, which can be executed concurrently across replicas. Second, on sparse graphs, independent sets admit parallelism via chromatic sampling. Consequently, high-throughput implementations are possible using CPUs, GPUs, FPGAs, custom accelerators or nanodevices. Our FPGA design is one point in this space and was chosen to accelerate the experiments in this work. The wall-clock numbers in Table II are an implementation case study rather than a universal hardware comparison.

All simulations use a uniform initialization, implemented by prepending a $\beta=0$ layer to the schedule. This yields random initial conditions and ensures randomized spin states for each replica. Further hardware design details are provided in the supplementary material (Sec. 4).

VI. PAOA VERSUS QAOA: SHERRINGTON-KIRKPATRICK MODEL

The Sherrington-Kirkpatrick (SK) model defines a mean-field spin glass with all-to-all random couplings, and serves as a canonical benchmark for optimization algorithms. The classical energy function is defined as

$$E(\{m\}) = -\frac{1}{\sqrt{n}} \frac{1}{\sqrt{N}} \sum_{i < j} J_{ij} m_i m_j \quad (7)$$

where $m_i \in \{-1, 1\}$ and $J_{ij} \sim \mathcal{N}(0, 1)$. The exact ground-state energy per spin in the large- n limit is known [29], but efficient algorithms for solving individual instances remain an open challenge.

QAOA has been studied extensively on the SK model [8, 16, 27, 30], with recent numerical work suggesting it can approximate the ground state in the infinite-size limit [31]. To enable a direct comparison, we evaluate PAOA using a two-schedule ansatz with exactly $2p$ parameters, matching the parameter count of depth- p QAOA. ~~Even though PAOA can easily be scaled up to very large n .~~ Although PAOA scales readily to much larger N , we stick to a fixed size of $nN = 26$, a typically studied SK-model, to enable a direct comparison with QAOA.

Each schedule is assigned to one half of the graph, split ~~arbitrarily.~~ For n uniformly at random. For $N = 26$, we randomly generate 30 training instances and optimize PAOA separately for each instance and layer depth $p \in \{1, \dots, 17\}$. The resulting schedules are then averaged and applied without further adjustment to a disjoint set of 30 test instances. QAOA results plotted in FIG. 4a use optimal parameters from prior work [8, 27]. For each instance, exact ground states are computed by exhaustive search. ~~PAOA with multiple schedules on the SK model with Lévy bonds. (a) Coupling strengths λ_i for each node, sorted in descending order for an $n = 50$ instance. The heavy-tailed distribution creates a clear separation between “heavy” and “light” nodes, as shown schematically in the inset. (b, c) Example schedules for the $n = 50$ benchmark with $p = 17$ layers. Panel (b) shows the two-schedule ansatz, while (c) shows the multi-schedule ansatz with 10 private schedules for the heavy nodes. (d) The corresponding schedule setups for the $n = 500$ benchmark. The inset shows the two-schedule ansatz, while the main plot shows the 100-schedule version. (e) Average energy per spin comparing a single SA schedule (red), a two-schedule PAOA (blue), and a multi-schedule PAOA (green) for both $n = 50$ and $n = 500$. For both system sizes, increasing the number of schedules consistently improves performance, with the multi-schedule strategy yielding the lowest energies.~~

Our results show that PAOA has highly competitive performance under these iso-parametric conditions. As shown in FIG. 4a, the average energy per spin consistently decreases with layer depth, and schedules trained on $nN = 26$ generalize well to much larger instances up to $nN = 500$ (see FIG. S5). To formalize this comparison, we report the approximation ratio in FIG. 4b, defined as

$$\text{Approx. Ratio} = \frac{E(\{m\})}{E(\{m_{\text{sol}}\})} \quad (8)$$

where $E(\{m_{\text{sol}}\})$ is the cost of the exact ground state. PAOA achieves a consistently higher approximation ratio than standard QAOA across all depths. Notably, the performance of vanilla PAOA is comparable to that of recently proposed hybrid quantum-classical methods that use classical techniques to improve QAOA’s performance [32], suggesting that a definitive quantum advantage on this problem has yet to be firmly established.

We emphasize that ~~while QAOA may offer clear advantages on problems that rely explicitly on quantum interference (one clean example is a clear quantum advantage over classical algorithms is expected to emerge in problems where interference plays an explicit role, such as in the~~

~~synthetic examples constructed by Montanaro et al. [13]. In these cases, the superposition of many computational paths, enabled by either the problem structure or the quantum algorithm, creates interference patterns that amplify the correct solution while suppressing others, an effect that classical probabilistic samplers struggle to reproduce due to the well-known “sign problem” [33]. As shown in [13]–[34], a probabilistic formulation of generic unitary evolutions yields complex (often purely imaginary) weights, so sampling effectively becomes randomized. While post-phase corrections recover unbiased estimates, the number of samples grows exponentially.~~

~~These interference-dominated regimes are precisely where we do not expect PAOA, or any classical sampler, to compete. On the other hand, unless such interference-driven advantages are clearly identified and demonstrated more broadly across practical problem classes, we do not expect any significant quantum advantage from QAOA. Meanwhile, such effects must be demonstrated on a case-by-case basis. Until then, PAOA offers a practical baseline rigorous, scalable and high-performing benchmark for variational sampling at problem sizes well beyond the reach of current quantum devices.~~

VII. MULTI-SCHEDULE ANNEALING IN THE LÉVY SK MODEL

~~The SK model contains~~ To a heuristic solver, the SK model presents no intrinsic structure beyond a dense random coupling graph. To explore whether PAOA can adaptively exploit structure when it ~~exists~~ might exist, we consider a variant where the coupling weights J_{ij} follow a heavy-tailed Lévy distribution [35]:

$$P(J) = \frac{\alpha}{2} |J|^{-1-\alpha}, \quad |J| > 1 \quad (9)$$

We use $\alpha = 0.9$, which produces a distribution with diverging variance, dominated by rare, large couplings.

In this regime, a natural question arises: can PAOA discover ~~heuristics~~ principles that treat strongly and weakly coupled nodes differently? Prior work on Boltzmann machines and annealing schedules [36]–[36, 37] suggests that high-degree or strongly coupled nodes benefit from slower annealing (more time for high temperature) to avoid freezing too early. ~~To test this idea, we assign distinct schedules to “heavy” and “light” nodes, defined by the per-node coupling strength. In these prior works, the heuristics are hand-designed by the ingenuity of the human algorithm designers. Here, we test whether the principle of annealing subgraphs with different temperatures can be discovered by PAOA through black-box optimization.~~

~~We begin by introducing a coupling strength per node:~~

$$\lambda_i = \sum_j |J_{ij}| \quad (10)$$

~~Nodes are sorted~~ We then sort nodes by λ_i and split into a “heavy” group (top 20%) and a “light” group (bottom 80%), as ~~depicted~~ shown in FIG. 5a. The split here is arbitrary

FIG. 5. Learning a variational principle using a PAOA double-schedule ansatz. (a) Heavy-tailed SK: per-node coupling strengths for an $N=50$ instance, sorted; the heavy-tailed distribution separates “heavy” and “light” nodes. (b) PAOA training with two schedules ($2p$ parameters) assigned to heavy and light nodes based on their coupling strengths, initialized from an optimized single annealing schedule (cyan). Curves are averaged across 50 instances; red/blue denote heavy/light nodes. (c) Extrapolated double schedule suggested by (b): the heavy-node schedule is extended to more layers and the light-node schedule is scaled up following PAOA’s guidance. (d) Average success probability over 500 instances comparing single-schedule SA (blue) and double-schedule PAOA (red) for $N=50$. Error bars are 95% confidence intervals from 10^5 bootstrap samples. (e) Success probability showing per-instance comparison between PAOA ($2p$) and SA (p) at $p = 5 \times 10^5$.

and we confirmed that a 50%-50% split following the same procedure yields similar results (not shown).

To test if PAOA can leverage this structure, we compare the standard first identify a near-optimal single-schedule SA against two structured PAOA ansätze: a two-schedule version and a multi-schedule version where each heavy node receives a private schedule. The specific methodology for generating and selecting these schedules is detailed using simulated annealing at shallow depth ($p=17$). We then let COBYLA optimize two-schedules, corresponding to heavy and light nodes, freely. After convergence, we average the schedules across all instances. As shown in FIG. 5b, the optimized schedules exhibit a qualitative separation: heavy nodes are assigned a higher-temperature (lower β) profile, and light nodes anneal faster. We verified that the different schedules for heavy and light nodes lead to higher success probability for the double-schedule over the single-schedule. Due to the shallow depth ($p=17$) however the differences are not appreciably large.

However, extrapolating the double-schedules discovered by PAOA to much deeper layers (FIG. 5c), we observe a clear difference against single-schedule SA. Our extrapolation is relatively straightforward but we report the details in the Methods section. The results, shown in FIG. 5d

for $N \in \{50, 500\}$ and for $N=50$ and 500 unseen instances, show that the two-schedule ansatz outperforms the single schedule, and the multi-schedule variant yields the lowest energies with a clear gap in success probabilities between the two approaches, confirming a robust benefit from structured, heterogeneous annealing. Details on determining the success probability for single and double schedules are in the Methods section.

We then ask whether PAOA itself can learn this strategy. Using the same instances, we apply a two-schedule parameterization and allow COBYLA to optimize both schedules freely. After convergence, we average the schedules across all instances. As shown in FIG. ??, the optimized schedules exhibit the same qualitative separation: heavy nodes are assigned a higher-temperature (lower β) profile, and light nodes anneal faster. This behavior suggests that PAOA can discover problem-aware heuristics automatically computationally. While such adaptive schedules have long been used manually in structured settings [38], we are not aware of prior variational algorithms for non-equilibrium sampling that systematically learn them from data. In this sense, PAOA serves as a tool for discovering new algorithmic strategies in disordered systems.

VIII. CONCLUSION

We have introduced the Probabilistic Approximate Optimization Algorithm (PAOA), a variational Monte Carlo framework that generalizes simulated annealing and supports a wide range of parameterizations compatible with existing Ising hardware. By treating the energy landscape itself as a variational object and updating it through classical feedback from Monte Carlo samples, PAOA enables efficient sampling from non-equilibrium distributions using shallow Markov chains.

We demonstrated that PAOA captures simulated annealing as a limiting case and, without constraints, can rediscover SA-like behavior when optimized for energy minimization. Using a minimal global schedule ansatz, PAOA efficiently learned monotonic cooling profiles that converge to low-energy states on 3D spin-glass problems. The approach was implemented on an FPGA-based p-computer, achieving hardware-accelerated annealing at rates exceeding CPU-based implementations by several orders of magnitude.

Beyond global schedules, we explored richer ansätze with multiple temperature profiles. On the ~~Sherrington-Kirkpatrick~~—Sherrington-Kirkpatrick (SK) model, PAOA ~~matched~~exceeded the performance of QAOA under iso-parametric conditions, while scaling to system sizes beyond those accessible to quantum devices. In a heavy-tailed variant of the SK model, PAOA automatically learned to assign slower annealing schedules to strongly coupled nodes, reproducing hand-crafted heuristics from prior work. As such, PAOA can be a tool for discovering new algorithmic strategies for sampling in disordered systems.

METHODS

Numerical simulation uses double precision (64-bit) in C++ to generate results for PAOA in FIG. 2d, FIG. 4a, and FIG. 5d. QAOA results were reproduced from [39].

1. Data transfer between FPGA and CPU

A PCIe interface was used to communicate between FPGA and CPU through a MATLAB interface for the ‘read/write’ operations. A global ‘disable/enable’ signal broadcast from MATLAB to the FPGA was used to freeze and resume all p-bits. Before a ‘read’ instruction, the p-bit states were saved to the local block memory (BRAM) with a snapshot signal. Then the data were read once from the BRAM using the PCIe interface and sent to MATLAB for post-processing, that is, updating the annealing schedule ($\beta^{(k)}$). For the ‘write’ instruction, the ‘disable’ signal was sent from MATLAB to freeze the p-bits before sending the updated schedule. After the ‘write’ instruction was given, p-bits were enabled again with the ‘enable’ signal sent from MATLAB.

2. Measurement of wall-clock time and flips per nanosecond

To measure the annealing wall-clock time per replica in FPGA along with the reading and writing overhead, we use MATLAB’s built-in ‘tic’ and ‘toc’ functions and average the total annealing time over 100 independent experiments.

Adding all flip attempts for the entire system with all replicas, 2160 p-bits, the total flips per nanosecond (flips/ns) are computed. For CPU measurements, MATLAB’s built-in ‘tic’ and ‘toc’ functions were used to measure the total annealing time taken to perform the MCS in Table III with 1000 runs each. The average time per MCS is reported for a single replica for a network size of $L^3 = 6^3$.

3. Schedule Generation for the Lévy SK Model Benchmark

The performance benchmark in Sec. VII required comparing ~~three~~two annealing strategies. The schedules for these strategies were generated via a grid search based on a geometric functional form:

$$\beta^{(k)} = \beta_{\text{initial}} \cdot \left(\frac{\beta_{\text{final}}}{\beta_{\text{initial}}} \right)^{k/(p-1)}, \quad k = 0, \dots, p-1 \quad (11)$$

The parameter ranges searched were $\beta \in [0.005, 0.12]$ for the $N = 50$ instances, ~~and $\beta \in [0.0001, 0.011]$ for the $N = 500$ instances.~~

The ~~three~~two strategies were constructed as follows:

- **Single-Schedule (SA):** We performed a grid search over β_{final} to find the single geometric schedule that produced the lowest average energy across all instances. This served as the baseline.
- **Two-Schedule PAOA:** The optimized baseline schedule was assigned to the ‘heavy’ nodes. ~~The schedule for the ‘’, as suggested by PAOA. For ‘light’ nodes was a scaled version of this baseline, where the optimal scaling factor was found via a separate grid search.~~
- **Multi-Schedule PAOA:** ~~The baseline schedule was assigned to the single heaviest node. Schedules for the other heavy nodes were generated recursively, with each subsequent node’s schedule scaled by a factor determined through grid search. All ‘light’ nodes shared a final, common schedule.~~ we scale up the baseline schedule by a scaling factor that reflects the separation proposed by PAOA. Then, both geometric schedules are extrapolated to large number of layers to study the effect of deep PAOA.

In order to compare the two approaches, we define success probability as the ratio of the number of runs that hit the putative ground state to the total number of runs (i.e., 100 independent runs). The putative ground state was found by running a long simulated annealing with one million sweeps and ten independent runs; then choosing the lowest energy among all ten million states.

4. Optimizer choice

For the outer optimization loop, we used the COBYLA algorithm as it is a robust and widely used method for derivative-free optimization problems with a moderate number of variables [40–42]. Our COBYLA implementation is based on the the non-linear optimization package (Nlopt)

[43]. While the performance of PAOA may depend on the choice of classical optimizer, a detailed comparison of different optimization techniques is beyond the scope of this work.

ACKNOWLEDGEMENTS

ASA, SC, and KYC acknowledge support from the National Science Foundation (NSF) under award number 2311295–2311295, and the Office of Naval Research (ONR), Multidisciplinary University Research Initiative (MURI) under Grant No. N000142312708. We are grateful to Navid Anjum Aadit for discussions related to the hardware implementation of online annealing, and Ruslan Shaydulin and Zichang He for input on QAOA benchmarking. Use was made of computational facilities purchased with funds from the National Science Foundation (CNS-1725797) and administered by the Center for Scientific Computing (CSC). The CSC is supported by the California NanoSystems Institute and the Materials Research Science and Engineering Center (MRSEC; NSF DMR 2308708) at UC Santa Barbara.

DATA AND CODE AVAILABILITY

All data and code used to generate the results and plots in this study are openly accessible at the following Github repository: <https://github.com/OPUSLab/PAOAwithPbits>.

AUTHOR CONTRIBUTIONS

KYC and ASA conceived the study. ASA and SC led the implementation of PAOA, including all simulations and hardware experiments. FM provided key insights into the theoretical framework and the interpretation of spin-glass benchmarks. All authors contributed to analyzing the results and writing the manuscript.

COMPETING INTERESTS

The authors declare no competing interests.

REFERENCES

- [1] G. Weitz, L. Pira, C. Ferrie, and J. Combes, arXiv preprint arXiv:2308.14981 (2023) *Physical Review A* **112**, 032418 (2025).
- [2] W. Wang, J. Machta, and H. G. Katzgraber, *Physical Review E* **92**, 013303 (2015).
- [3] Z.-S. Shen, F. Pan, Y. Wang, Y.-D. Men, W.-B. Xu, M.-H. Yung, and P. Zhang, *Nature Computational Science* **5**, 322 (2025).
- [4] M. Muñoz-Arias, M. Y. Niu, W. W. Ho, A. Rattew, M. Broughton, and M. Mohseni, *Phys. Rev. Res.* **6**, 023294 (2024).
- [5] Q. Ma, Z. Ma, J. Xu, H. Zhang, and M. Gao, *Communications Physics* **7**, 236 (2024).
- [6] A. Barzegar, F. Hamze, C. Amey, and J. Machta, *Physical Review E* **109**, 065301 (2024).
- [7] C. Fan, M. Shen, Z. Nussinov, Z. Liu, Y. Sun, and Y.-Y. Liu, *Nature Communications* **14**, 725 (2023).
- [8] J. Basso, E. Farhi, K. Marwaha, B. Villalonga, and L. Zhou (Schloss Dagstuhl – Leibniz-Zentrum für Informatik, 2022).

- [9] R. Herrman, P. C. Lotshaw, J. Ostrowski, T. S. Humble, and G. Siopsis, *Scientific Reports* **12**, 6781 (2022).
- [10] V. Vijendran, A. Das, D. E. Koh, S. M. Assad, and P. K. Lam, *Quantum Science* **025010** (2024).
- [11] L. Zhou, S.-T. Wang, S. Choi, H. Pichler, and M. D. Lukin, *Physical Review X* **10** (2020).
- [12] T. Müller, A. Singh, F. K. Wilhelm, and T. Bode, *Phys. Rev. Res.* **7**, 023165 (2025).
- [13] A. Montanaro and L. Zhou, *arXiv preprint arXiv:2411.04979* (2024).
- [14] L. Zhou, S.-T. Wang, S. Choi, H. Pichler, and M. D. Lukin, *Physical Review X* **10**, 021067 (2020).
- [15] S. Boulebnane, J. Sud, R. Shaydulin, and M. Pistoia, *arXiv preprint arXiv:2503.09563* (2025).
- [16] B. Gülbahar, *IEEE Transactions on Wireless Communications* **23**, 11567 (2024).
- [17] J. R. Finžgar, M. J. Schuetz, J. K. Brubaker, H. Nishimori, and H. G. Katzgraber, *Physical Review Research* **6**, 023063 (2024).
- [18] L. Cheng, Y.-Q. Chen, S.-X. Zhang, and S. Zhang, *Communications Physics* **7**, 83 (2024).
- [19] K. Y. Çamsarı, B. M. Sutton, and S. Datta, *Applied Physics Reviews* **6**, 011305 (2019).
- [20] W. A. Borders, A. Z. Pervaiz, S. Fukami, K. Y. Çamsarı, H. Ohno, and S. Datta, *Nature* **573**, 390 (2019).
- [21] N. A. Aadit, A. Grimaldi, M. Carpentieri, L. Theogarajan, J. M. Martinis, G. Finocchio, and K. Y. Camsari, *Nature Electronics* **5**, 460 (2022) N. A. Aadit, S. Nikhar, S. Kannan, S. Chowdhury, and K. Y. Çamsarı, *Nature Communications* **15**, 53270–8977 (2024).
- [22] S. Nikhar, S. Kannan, N. A. Aadit, S. Chowdhury, and K. Y. Camsari, *Nature Communications* **15**, 53270–8977 (2024).
- [23] S. Chowdhury and K. Y. Çamsarı, *arXiv preprint arXiv:2310.06679* (2023).
- [24] G. Carleo and M. Troyer, *Science* **355**, 602 (2017).
- [25] M. J. D. Powell, in *Advances in Optimization and Numerical Analysis*, edited by S. Gomez and J.-P. Hennart (Springer, 1994) pp. 51–67.
- [26] G. E. Hinton, *Neural computation* **14**, 1771 (2002).
- [27] E. Farhi, J. Goldstone, S. Gutmann, and L. Zhou, *Quantum* **6**, 735 (2022).
- [28] P. C. Lotshaw, G. Siopsis, J. Ostrowski, R. Herrman, R. Alam, S. Powers, and 042411 (2023).
- [29] G. Parisi, *Physical Review Letters* **43**, 1754 (1979).
- [30] A. Crisanti and T. Rizzo, *Physical Review E* **65**, 046137 (2002).
- [31] S. Boulebnane, A. Khan, M. Liu, J. Larson, D. Herman, R. Shaydulin, and M. Pistoia, *arXiv preprint arXiv:2505.07929* (2025).
- [32] M. Dupont and B. Sundar, *Phys. Rev. A* **109**, 012429 (2024).
- [33] M. Troyer and U.-J. Wiese, *Physical review letters* **94**, 170201 (2005).
- [34] S. Chowdhury, K. Y. Çamsari, and S. Datta, *IEEE Access* **11**, 116944 (2023).
- [35] S. Boettcher, *Philosophical Magazine* **92**, 34 (2012) E. Aarts, J. Korst, and W. Michiels, -.
- [36] E. Aarts and J. Korst, in *Search methodologies: introductory tutorials in optim (2005) Simulated Annealing and Boltzmann Machines: A Stochastic Approach (John Wiley & Sons, 1989) Chap. 8, pp. 150–151.*
- [37] J. I. Adame and P. L. McMahon, *Quantum Science and Technology* **5**, 035011 (2020).
- [38] M. Mohseni, D. Eppens, J. Strumpfer, R. Marino, V. Denchev, A. K. Ho, S. V. Isakov, S. Boixo, F. Ricci-Tersenghi, and H. Neven, *arXiv preprint arXiv:2111.13628* (2021).
- [39] D. Lykov, R. Shaydulin, Y. Sun, Y. Alexeev, and M. Pistoia, in *Proceedings of the SC'23 Workshops of The International Conference on High Performance Computing, Network,*

Storage, and Analysis (2023) pp. 1443–1451.

- [40] A. Pellow-Jarman, I. Sinayskiy, A. Pillay, and F. Petruccione, *Quantum Information Processing* **20** (2021).
- [41] A. Arrasmith, M. Cerezo, P. Czarnik, L. Cincio, and P. J. Coles, *Quantum* **5**, 558 (2021).
- [42] B. F. Schiffer, J. Tura, and J. I. Cirac, *PRX Quantum* **3**, 020347 (2022).
- [43] S. G. Johnson, The NLOpt nonlinear-optimization package (2007).
- [44] D. Blackman and S. Vigna, ~~Scrambled linear pseudorandom number generators~~ ACM Trans. Math. Softw. **47** (2021).

Supplementary Information

Probabilistic Approximate Optimization: A New Variational Monte Carlo Algorithm

Abdelrahman S. Abdelrahman, Shuvro Chowdhury, Flaviano Morone and Kerem Y. Camsari

Here, we present the PAOA fully parameterized algorithm, which is used in learning the full-adder target states, see section. 1. The exact same structure of the algorithm is also used for learning the majority gate correct states, however, with private β as ansatz, see section. 2. The p-computer subroutine uses p-bit equations to generate samples from a distribution, defined by the

Algorithm 2: PAOA: fully-parameterized

Input : number of nodes N , number of layers p , number of experiments N_E , initial variational parameters $(J^{(1)}, J^{(2)}, \dots, J^{(p)})$, tolerance $\varepsilon_{\text{step}}$, maximum iterations t_{max} , truth table

Output: trained set of weights $(J_{\text{opt}}^{(1)}, J_{\text{opt}}^{(2)}, \dots, J_{\text{opt}}^{(p)})$

```

1 Function p-computer ( $J_{\text{init}}, N, p$ ):
2   initialize all spins randomly
3   for  $i \leftarrow 1$  to  $p$  do
4      $J \leftarrow J^{(i)}$ 
5     for  $j \leftarrow 1$  to  $N$  do
6       solve Eqs. \(S.1\) and \(S.2\)
7   return  $p$ -bit states in decimal
8 Function PAOA-circuit ( $N_E, J, N, p$ ):
9   for  $k \leftarrow 1$  to  $N_E$  do
10    state  $\leftarrow$  p-computer ( $J, N, p$ )
11    save the p-bit states
12    find the estimated distribution ( $\hat{\rho}_f$ ) using equation \(S.3\)
13    compute the cost ( $\mathcal{L}$ ) using equation \(S.4\)
14    return cost
15 while ( $\text{step size} > \varepsilon_{\text{step}}$  and  $\text{number of iterations} < t_{\text{max}}$ ) do
16   cost  $\leftarrow$  PAOA-circuit( $N_E, J, N, p$ )
17   minimize cost and get a perturbation vector ( $p$ ) using a gradient-free optimizer
18   for  $i \leftarrow 1$  to  $p$  do
19      $J_{t+1}^{(i)} \leftarrow J_t^{(i)} + p^{(i)}$ 
20    $t \leftarrow t + 1$ 
21 return optimal variational parameters

```

updated (J, h) . In equations. (S.1) and (S.2), we used two variations of this algorithm. In the full-adder case, we set $\beta = 1, h = 0$, and optimize J .

$$m_i = \text{sgn}[\tanh(\beta I_i) - \text{rand}_u(-1, 1)], \quad (\text{S.1})$$

In the majority gate problem, the graph weights and biases are set to one and zero, respectively ($J_{ij} = +1, h = 0$). Then, the annealing schedule is localized for each node (β_i), that is, each p-bit gets its own schedule.

$$I_i = \sum_j J_{ij} m_j + h_i. \quad (\text{S.2})$$

Since the cost function is defined over the distribution of the target states, we estimated the distribution using the generated independent samples by counting the frequency of observing the desired states over N_E independent experiments (Eq. S.3). For accurate estimation, we use ten million experiments, which bounds the deviation from the true distribution to approximately 3.2×10^{-4} .

$$\hat{\rho}^{(j)}(\{m\}) = \frac{1}{N_E} \sum_{k=1}^{N_E} \mathbb{1}\{X_{j,k} = \{m\}\}, \quad \{m\} \in \Omega \quad (\text{S.3})$$

In the negative log-likelihood function defined in Eq. (S.4), the estimated distribution over a target states set \mathcal{X} is used to calculate the associated loss. The variational parameters $\underline{\theta}$ represent the parameters being optimized, such as J_{ij} for the full-adder, and β_i for the majority gate problem.

$$\mathcal{L}(\underline{\theta}) = - \sum_{\{m\} \in \mathcal{X}} \ln(\hat{\rho}_k(\{m\}; \underline{\theta})), \quad (\text{S.4})$$

1. Full-Adder with Fully Parametrized PAOA Ansatz

FIG. S1. (a) All-to-All full-adder network, showing graph weights $J(p)$, and truth table, where Dec. refers to the decimal representation of the state of [A B C_{in} S C_{out}] from left to right. (b) Training loss over optimization iterations for gradient-based and gradient-free methods. (c) PDF evolution across three layers, obtained from a Markov chain. (d) PDF evolution across three time layers, using MCMC (10^7 experiments).

The full-adder is represented as a fully connected network comprising five p-bits, where the graph weights are the only variational parameters (FIG. S1a). As depicted in the same figure, the graph weights evolve with p , such that for each layer, we have different weights that are responsible for the PDF evolution. The full-adder performs 1-bit binary addition with three inputs (A, B, and Carry in = C_{in}) and two outputs (Sum = S and Carry out = C_{out}). We use a finite Markov chain with three layers ($p = 3$), and $\rho_0 = [1 \ 0 \ 0 \ 0 \ 0]^T$ as the initial condition PDF. The optimization is carried out exactly similar to the majority gate problem in section. 2, except the parameterization here is the graph weights. The optimal parameters, rounded to the nearest hundredth, are presented in Eq. (S.5). The results in FIG. S1c and FIG. S1d show clear agreement between the two methods, namely, the gradient-based approach where the probability transition matrix is constructed, and the gradient-free one where samples are generated to estimate the underlying distribution. More details on the gradient-based method are provided in section. 3, where we show how an AND gate can be solved analytically.

$$J_{\text{optimal}}^{(1)} = \begin{pmatrix} 0 & 0.18 & 0.12 & 0 & -0.10 \\ 0.18 & 0 & -0.08 & 0.06 & -0.11 \\ 0.12 & -0.08 & 0 & 0.01 & -0.05 \\ 0 & 0.06 & 0.01 & 0 & 0.02 \\ -0.10 & -0.11 & -0.05 & 0.02 & 0 \end{pmatrix}, J_{\text{optimal}}^{(2)} = \begin{pmatrix} 0 & 0.31 & 0.43 & 1.14 & 0.17 \\ 0.31 & 0 & 0.15 & 0.58 & -0.32 \\ 0.43 & 0.15 & 0 & 1.26 & -0.28 \\ 1.14 & 0.58 & 1.26 & 0 & 1.45 \\ 0.17 & -0.32 & -0.28 & 1.45 & 0 \end{pmatrix}, J_{\text{optimal}}^{(3)} = \begin{pmatrix} 0 & -1.68 & -1.91 & 1.90 & 1.93 \\ -1.68 & 0 & -2.31 & 1.87 & 2.25 \\ -1.91 & -2.31 & 0 & 1.84 & 2.58 \\ 1.90 & 1.87 & 1.84 & 0 & -3.93 \\ 1.93 & 2.25 & 2.58 & -3.93 & 0 \end{pmatrix} \quad (\text{S.5})$$

The simulation parameters used here are tabulated in Table S1. In gradient-based approach, the training terminates when either the tolerance in gradient or maximum iterations is met, $\|\nabla\mathcal{L}(\boldsymbol{\theta})\| < \varepsilon_{\text{grad}}$, because updates of such order no longer improve the objective. For derivative-free ‘‘COBYLA’’, convergence is declared when the step-size tolerance (trust–region radius) falls below $\Delta\theta < \varepsilon_{\text{step}}$, signalling that all admissible simplex moves would alter the variational parameters by less than numerical precision, and hence further iterations are unproductive.

TABLE S1. Full Adder Simulation Parameters.

Parameter	Value
number of nodes (N)	5
number of layers (p)	3
update order	$\{m_1, m_2, m_3, m_4, m_5\}$
initial parameters ($J^{(1)}, J^{(2)}, J^{(3)}$)	0.1
learning rate (η)	0.01
tolerance in gradient ($\varepsilon_{\text{grad}}$)	10^{-6}
maximum iterations ($\mathcal{M}t_{\text{max}}$)	2000
number of experiments (N_E)	10^7
tolerance ($\varepsilon_{\text{step}}$)	10^{-6}

2. Majority Gate Problem

In this section, we present the optimal parameters obtained for the majority gate problem using a local-annealing schedule ansätze with two layers ($p=2$). In this formulation, each node is assigned a private inverse temperature β , allowing the model to adaptively capture local structure in the optimization landscape. The full set of simulation parameters used to obtain these results is summarized in Table S2.

TABLE S2. Majority Gate Simulation Parameters.

Parameter	Value
number of nodes (N)	4
number of layers (p)	2
update order	$\{m_1, m_2, m_3, m_4\}$
initial parameters ($\bar{J}^{(1)}, \bar{J}^{(2)}$)	1
learning rate (η)	0.004
tolerance in gradient ($\varepsilon_{\text{grad}}$)	10^{-7}
maximum iterations ($\mathcal{M}t_{\text{max}}$)	5000
number of experiments (N_E)	10^7
tolerance ($\varepsilon_{\text{step}}$)	10^{-7}

The optimal parameters for the nodes labeled $[A, B, C, Y]$, rounded to the nearest decimal, are:

$$\beta_{\text{node}} \begin{matrix} p=1 & p=2 \\ \beta_A \begin{bmatrix} 0.8 & -0.2 \\ 0.8 & 0 \\ 0.9 & 0 \\ 0.5 & 2.7 \end{bmatrix} \end{matrix} \quad (\text{S.6})$$

These variational parameters, when combined with the initial graph weights ($J_{ij}=+1$), define the effective couplings $\bar{J}_{ij}^{(k)} = \beta_i^{(k)} J_{ij}$. The optimized parameters can be directly used to reproduce the correct peaks of the majority gate.

3. Analytical Formulation of PAOA: AND Gate Problem

FIG. S2. (a) A schematic of AND gate as an undirected graph. (b) The AND gate schematic along with the truth table, where Dec. refers to the decimal representation of the state $[m_1, m_2, m_3]$ from left to right.

In this section, we show the details of learning the weights of an AND gate analytically using the fully parameterized PAOA ansatz. The problem is shown in FIG. S2. Using the *fully-parameterized-PAOA*, the J graph weight matrix and bias vector h can be written as:

$$J = \begin{pmatrix} 0 & J_{12} & J_{13} \\ J_{12} & 0 & J_{23} \\ J_{13} & J_{23} & 0 \end{pmatrix}, \quad h = \begin{bmatrix} h_1 \\ h_2 \\ h_3 \end{bmatrix} \quad (\text{S.7})$$

For a general graph, the entries of w_k are given by

$$[w_k]_{ab} = P(a \leftarrow b) = \begin{cases} \frac{1 + m_k^{(a)} \tanh(I_k^{(b)})}{2} & \text{if } m^{(a)} \text{ and } m^{(b)} \text{ are identical} \\ 0 & \text{except possibly at the } k^{\text{th}} \text{ bit,} \\ & \text{otherwise} \end{cases} \quad (\text{S.8})$$

where $I_k^{(b)} = \sum_{\ell} J_{k\ell} m_{\ell}^{(b)} + h_k$ is the synaptic input to node k based on the configuration $m^{(b)}$, and $m^{(a)}$ denotes the spin configuration after updating bit k .

Using equations (S.7) and (S.8) with the following update order $\{m_1, m_2, m_3\}$, the probability transition matrix (W) can be constructed as follows:

$$W = w_3 w_2 w_1 = \underbrace{\begin{bmatrix} t & t & 0 & 0 & 0 & 0 & 0 & 0 \\ t' & t' & 0 & 0 & 0 & 0 & 0 & 0 \\ 0 & 0 & u & u & 0 & 0 & 0 & 0 \\ 0 & 0 & u' & u' & 0 & 0 & 0 & 0 \\ 0 & 0 & 0 & 0 & u' & u' & 0 & 0 \\ 0 & 0 & 0 & 0 & u & u & 0 & 0 \\ 0 & 0 & 0 & 0 & 0 & 0 & t' & t' \\ 0 & 0 & 0 & 0 & 0 & 0 & t & t \end{bmatrix}}_{\text{update } m_3 \text{ conditioned on } \{m_2, m_1\}} \times \underbrace{\begin{bmatrix} r & 0 & r & 0 & 0 & 0 & 0 & 0 \\ 0 & s & 0 & s & 0 & 0 & 0 & 0 \\ r' & 0 & r' & 0 & 0 & 0 & 0 & 0 \\ 0 & s' & 0 & s' & 0 & 0 & 0 & 0 \\ 0 & 0 & 0 & 0 & s' & 0 & s' & 0 \\ 0 & 0 & 0 & 0 & 0 & r' & 0 & r' \\ 0 & 0 & 0 & 0 & s & 0 & s & 0 \\ 0 & 0 & 0 & 0 & 0 & r & 0 & r \end{bmatrix}}_{\text{update } m_2 \text{ conditioned on } \{m_3, m_1\}} \times \underbrace{\begin{bmatrix} p & 0 & 0 & 0 & p & 0 & 0 & 0 \\ 0 & q & 0 & 0 & 0 & q & 0 & 0 \\ 0 & 0 & q' & 0 & 0 & 0 & q' & 0 \\ 0 & 0 & 0 & p' & 0 & 0 & 0 & p' \\ p' & 0 & 0 & 0 & p' & 0 & 0 & 0 \\ 0 & q' & 0 & 0 & 0 & q' & 0 & 0 \\ 0 & 0 & q & 0 & 0 & q & 0 & 0 \\ 0 & 0 & 0 & p & 0 & 0 & 0 & p \end{bmatrix}}_{\text{update } m_1 \text{ conditioned on } \{m_3, m_2\}} \quad (\text{S.9})$$

$$= \begin{matrix} & 000 & 001 & 010 & 011 & 100 & 101 & 110 & 111 \\ \begin{matrix} 000 \\ 001 \\ 010 \\ 011 \\ 100 \\ 101 \\ 110 \\ 111 \end{matrix} & \begin{bmatrix} prt & qst & q'rt & p'st & prt & qst & q'rt & p'st \\ prt' & qst' & q'rt' & p'st' & prt' & qst' & q'rt' & p'st' \\ pr'u & qs'u & q'r'u & p's'u & pr'u & qs'u & q'r'u & p's'u \\ pr'u' & qs'u' & q'r'u' & p's'u' & pr'u' & qs'u' & q'r'u' & p's'u' \\ p's'u' & q'r'u' & qs'u' & pr'u' & p's'u' & q'r'u' & qs'u' & pr'u' \\ p's'u & q'r'u & qs'u & pr'u & p's'u & q'r'u & qs'u & pr'u \\ p'st' & q'rt' & qst' & prt' & p'st' & q'rt' & qst' & prt' \\ p'st & q'rt & qst & prt & p'st & q'rt & qst & prt \end{bmatrix} \end{matrix}$$

where

$$\begin{aligned}
p &= \frac{1 + \tanh(J_{12} + J_{13} + h_1)}{2}, & q &= \frac{1 + \tanh(J_{12} - J_{13} + h_1)}{2}, & r &= \frac{1 + \tanh(J_{12} + J_{23} + h_2)}{2} \\
p' &= \frac{1 - \tanh(J_{12} + J_{13} + h_1)}{2}, & q' &= \frac{1 - \tanh(J_{12} - J_{13} + h_1)}{2}, & r' &= \frac{1 - \tanh(J_{12} + J_{23} + h_2)}{2} \\
s &= \frac{1 + \tanh(J_{12} - J_{23} + h_2)}{2}, & t &= \frac{1 + \tanh(J_{13} + J_{23} + h_3)}{2}, & u &= \frac{1 + \tanh(J_{13} - J_{23} + h_3)}{2} \\
s' &= \frac{1 - \tanh(J_{12} - J_{23} + h_2)}{2}, & t' &= \frac{1 - \tanh(J_{13} + J_{23} + h_3)}{2}, & u' &= \frac{1 - \tanh(J_{13} - J_{23} + h_3)}{2}
\end{aligned} \tag{S.10}$$

Note that $x + x' = 1$, $x \in \{p, q, r, s, t, u\}$. To illustrate the procedure, let's examine the construction of following entry W_{81} . Using Gibbs sampling, we get

$$\begin{aligned}
W_{81} &= P(\{1, 1, 1\} \leftarrow \{-1, -1, -1\}) \\
&= \underbrace{\frac{1 + \tanh(J_{12}m_2 + J_{13}m_3 + h_1)}{2}}_{m_2=-1, m_3=-1} \times \underbrace{\frac{1 + \tanh(J_{21}m_1 + J_{23}m_3 + h_2)}{2}}_{m_1=1, m_3=-1} \times \underbrace{\frac{1 + \tanh(J_{31}m_1 + J_{32}m_2 + h_3)}{2}}_{m_1=1, m_2=1} \\
&= p'st
\end{aligned} \tag{S.11}$$

Note the use of updated value of m_1 in updating m_2 . Similarly, we used the updated value of m_1 and m_2 in updating m_3 . This is basically indicating that p-bits are updated sequentially. Following the same procedure, the rest of W entries can be filled. This W matrix is used now to train the AND gate with analytical derivatives.

We illustrate the procedure for training an AND gate, which can be extended in the same manner to a full-adder. The training is accomplished in three main steps:

- (i) *Construct the transition matrix.* We define the probability transition matrix W using the coupling weights $\{J_{12}, J_{23}, J_{13}\}$ and biases $\{h_1, h_2, h_3\}$, as depicted in FIG. S2(a). For a network with p layers, one may assign separate W (and hence distinct sets of J and h) for each layer.
- (ii) *Specify a loss function.* We adopt the loss function given by Eq. (5), evaluated over the ‘‘truth table’’ states in FIG. S2(b). For the AND gate, these states are $\mathcal{X} = \{(0, 0, 0), (0, 1, 0), (1, 0, 0), (1, 1, 1)\}$.
- (iii) *Perform gradient descent.* Using gradient-descent we iteratively update each weight and bias until convergence.

For simplicity, we set $p = 1$ (a single layer), which suffices to obtain the desired states. We also choose $\rho_0 = [1 \ 0 \ \dots \ 0]^\top$ as the initial configuration, though in principle any initial state may be used. Notably, the final optimal parameters will be valid only for this specific initial choice, although the approach generalizes to arbitrary ρ_0 .

Carrying out the above steps leads to

$$\underbrace{\begin{bmatrix} prt & qst & q'rt & p'st & prt & qst & q'rt & p'st \\ prt' & qst' & q'rt' & p'st' & prt' & qst' & q'rt' & p'st' \\ pr'u & qs'u & q'r'u & p's'u & pr'u & qs'u & q'r'u & p's'u \\ pr'u' & qs'u' & q'r'u' & p's'u' & pr'u' & qs'u' & q'r'u' & p's'u' \\ p's'u & q'r'u & qs'u & pr'u & p's'u & q'r'u & qs'u & pr'u \\ p's't & q'rt' & qst' & prt' & p's't & q'rt' & qst' & prt' \\ p'st & q'rt & qst & prt & p'st & q'rt & qst & prt \end{bmatrix}}_W \underbrace{\begin{bmatrix} 1 \\ 0 \\ 0 \\ 0 \\ 0 \\ 0 \\ 0 \\ 0 \end{bmatrix}}_{\rho_0} = \underbrace{\begin{bmatrix} prt \\ prt' \\ pr'u \\ pr'u' \\ p's'u \\ p's't \\ p'st \end{bmatrix}}_{\rho_f}, \tag{S.12}$$

where ρ_f then enters the loss function as follows:

$$\begin{aligned}
\mathcal{L}(\underline{\theta}) &= - \sum_{\{m\} \in \mathcal{X}} \ln[\rho_f(\{m\}; \underline{\theta})] \\
&= - [\ln(prt) + \ln(pr'u) + \ln(p's'u) + \ln(p'st)],
\end{aligned} \tag{S.13}$$

with parameter vector $\underline{\theta} = [J_{12} \ J_{23} \ J_{13} \ h_1 \ h_2 \ h_3]^\top$.

Its derivatives with respect to each parameter are given by:

$$\begin{aligned}
\frac{\partial \mathcal{L}}{\partial J_{12}} &= \operatorname{sech}(h_2 + J_{12}) \left\{ -\operatorname{sech}(h_1 + J_{12} + J_{13}) \left[\cosh(h_1 - h_2 + J_{13}) + \cosh(h_1 + h_2 + 2J_{12} + J_{13}) \right. \right. \\
&\quad \left. \left. - 4 \sinh(h_1 + h_2 + 2J_{12} + J_{13}) \right] + 2 \sinh(J_{23}) \left[-\operatorname{sech}(h_2 + J_{12} - J_{23}) + \operatorname{sech}(h_2 + J_{12} + J_{23}) \right] \right\} \\
\frac{\partial \mathcal{L}}{\partial J_{13}} &= 2 \operatorname{sech}(h_3 + J_{13}) \left\{ \sinh(J_{23}) (\operatorname{sech}[h_3 + J_{13} + J_{23}] - \operatorname{sech}[h_3 + J_{13} - J_{23}]) \right. \\
&\quad \left. - \operatorname{sech}(h_1 + J_{12} + J_{13}) (\cosh(h_1 - h_3 + J_{12}) + \cosh(h_1 + h_3 + J_{12} + 2J_{13}) - 2 \sinh(h_1 + h_3 + J_{12} + 2J_{13})) \right\} \\
\frac{\partial \mathcal{L}}{\partial J_{23}} &= -2 \tanh(h_2 + J_{12} - J_{23}) - 2 \tanh(h_3 + J_{13} - J_{23}) + 2 \tanh(h_2 + J_{12} + J_{23}) + 2 \tanh(h_3 + J_{13} + J_{23}) - 2 \\
&\quad \frac{\partial \mathcal{L}}{\partial h_1} = 4 \tanh(h_1 + J_{12} + J_{13}) - 2 \\
&\quad \frac{\partial \mathcal{L}}{\partial h_2} = 2 \operatorname{sech}(h_2 + J_{12} - J_{23}) \operatorname{sech}(h_2 + J_{12} + J_{23}) \sinh(2(h_2 + J_{12})) \\
\frac{\partial \mathcal{L}}{\partial h_3} &= -\operatorname{sech}(h_3 + J_{13} - J_{23}) \operatorname{sech}(h_3 + J_{13} + J_{23}) \left[\cosh(2(h_3 + J_{13})) + \cosh(2J_{23}) - 2 \sinh(2(h_3 + J_{13})) \right]
\end{aligned}$$

Finally, the gradient-descent update rules are:

$$\begin{aligned}
J_{ij}^{(t+1)} &\leftarrow J_{ij}^{(t)} - \eta \frac{\partial \mathcal{L}}{\partial J_{ij}^{(t)}}, \quad 1 \leq i < j \leq 3, \\
h_i^{(t+1)} &\leftarrow h_i^{(t)} - \eta \frac{\partial \mathcal{L}}{\partial h_i^{(t)}}, \quad 1 \leq i \leq 3,
\end{aligned} \tag{S.15}$$

where η is a hyperparameter (step size). Each iteration refines the parameters and thus updates the transition matrix W , yielding the new distribution $\rho_f^{(t+1)}$. The stopping criterion is set to be either a fixed iteration budget or a certain tolerance in the gradient.

Using the method explained above with $\eta = 0.02$, maximum iterations of 2000, tolerance in the gradient of 10^{-6} , and uniform random initialization of weights and biases, specifically, $\theta \sim \operatorname{rand}_u[-0.5, 0.5]$, we get the following rounded optimal weights and biases:

$$J_{\text{optimal}} = \begin{pmatrix} 0 & 0 & 2.25 \\ 0 & 0 & 2.25 \\ 2.25 & 2.25 & 0 \end{pmatrix}, \quad h_{\text{optimal}} = \begin{bmatrix} 2.25 \\ 2.25 \\ -2.25 \end{bmatrix} \tag{S.16}$$

The training loss across optimization iterations and the final distribution generated using the optimal parameters are shown in FIG. S2.

FIG. S3. (a) Cost versus optimization iterations for the probabilistic AND-gate, trained using the negative log-likelihood and gradient descent. (b) The system PDF evolution using the optimal weights and biases.

4. FPGA Implementation of On-chip Annealing

In the main paper, we present experimental results obtained using a hybrid classical-probabilistic computing system. Here, we provide the technical details of the FPGA-based p-computer implemented on the Xilinx VCU128 data center accelerator card.

FIG. S4. Probabilistic computer architecture with on-chip annealing. The **annealing unit** is a $p \times 1$ multiplexer controlled by a counter that updates β value after MCSs budget is elapsed. The **synapse** block implements Eq. (2) by using a 2×1 multiplexer and finding the sum over all states $\{m_j\}$. The **DSP** slice carries out the multiplication of β and the synapse input I_i . The **BSN** unit implements Eq. (1), uses lookup table for tanh, Xoshiro [44] as the pseudorandom-pseudorandom number generator, and a comparator to update the p-bit states.

The p-computer architecture with on-chip annealing (FIG. S4) consists of four main blocks as follows:

- **Annealing unit:** outputs β based on the MCS counter. The MCS counter increases once a fixed MCSs budget is elapsed. The value of the counter is then used to select the value of the β . After exhausting all β values for all layers, the counter value will start from the beginning for a new experiment. For the experiments to be independent, the initial β value is set to zero to randomize the p-bits. The fixed-point precision used here is $s[4][5]s[4][5]$, where “s” denotes the signed bit, and the values in the square brackets represent the integer and fraction bits, respectively.
- **Synapse:** outputs I_i based on the states, weights, and biases. For each m_j , the multiplexer either chooses J_{ij} or zero. All results are then added at the end along with the bias to calculate the input to node i . The fixed-point precision used here for weights and biases is $s[4][5]s[4][5]$.
- **DSP:** outputs the multiplication of I_i and β .
- **Binary stochastic neuron:** outputs the updated binary state of node i , m_i . The output of the DSP block is taken to the LUT to find the corresponding tanh value. This value is compared with a random number generated via Xoshiro [44] using a comparator.

5. Parameters Trained on Small-size SK Instances for Larger Problems

FIG. S5. The average energy per spin $\langle E \rangle / N$ for 30 random large SK instances of size $n \in \{400, 500\}$ $N \in \{400, 500\}$ using parameters trained on $n=26$ $N=26$ and 10^5 – 10^6 independent experiments.

In this section, we evaluate the generalization of parameters trained on $n=26$ $N=26$ -spin SK instances by applying them to significantly larger problems with $n=400$ and $n=500$ $N=400$ and $N=500$, without any additional fine-tuning. Figure S5 shows the average energy per spin for these larger Sherrington–Kirkpatrick systems, plotted alongside the Parisi value, which represents the ground state energy in the thermodynamic limit. As the number of PAOA layers increases, the average energy decreases and the instance-to-instance variation narrows and concentrates more so than in the $n=26$ $N=26$ case. This behavior suggests that the optimized schedules generalize effectively to larger system sizes, and that the remaining fluctuations are primarily due to finite-size effects. It is worth noting that problem sizes of this scale are currently intractable for existing QAOA hardware, underscoring a key practical advantage of PAOA.

6. Methodology of Schedule Privatization in the Lévy SK Model

To evaluate the performance difference between conventional single-schedule simulated annealing and a privatized-multi-schedule annealing scheme applied to the SK model with Lévy–Lévy bonds, the following two-step procedure is adopted:

- Baseline schedule optimization.** We perform a grid search over final inverse-temperature values to identify an optimal *single geometric annealing schedule*. The initial inverse temperature (corresponding to high temperature) is held fixed at a small value, while the final value is varied to minimize the average energy per spin across multiple *instances and multiple runs*.
- Privatized schedule generation.** Using the optimized schedule from Step 1, denoted as β_1 , we introduce schedule heterogeneity by applying a multiplicative spacing factor Δ , which is *determined via grid search*.
- ~~In the case of two schedules, the~~ *guided by the separation in PAOA.* The original β_1 is assigned to “heavy” nodes (i.e., those strongly coupled), while the “light” nodes receive a schedule scaled by $(1 + \Delta)$, i.e., $\beta_2 = (1 + \Delta)\beta_1$.
- ~~For the multi-schedule scenario, β_1 is assigned to the heaviest node (in terms of coupling strength), and subsequent node-specific schedules are defined recursively as $\beta_{i+1} = (1 + \Delta)\beta_i$, for $i \in \{1, \dots, N_{\text{heavy}} - 1\}$. The light nodes are assigned a common schedule equal to $\beta_{\text{light}} = (1 + \Delta)\beta_{N_{\text{heavy}}}$, where Δ is chosen to reflect the separation that shallow PAOA averaged schedule exhibited when trained on $N=50$ with 50 instances in Fig. 5b.~~

This procedure ~~ensures that all heavy nodes are annealed using successively slower schedules, while the remaining light nodes follow a shared, even slower schedule.~~ *follows closely the guidance of PAOA of scaling the optimized single schedule up rather than scaling down the best single annealing schedule and assigning it for the heavy nodes, while keeping the light nodes running at the best annealing schedule. This choice is determined solely by PAOA owing to its learning capability.*

The impact of this *privatized* scheduling strategy is assessed by comparing the *average energy per spin to the single-schedule baseline*.

7. PAOA-Discovered Schedule Separation in Lévy SK

The optimized two schedules for SK model with Lévy bonds using PAOA with $2p$ parameters averaged across 50 instances, where the initial schedule (green curve), which is $\exp(\text{linspace}(\log(0.005), \log(0.12), p = 17))$, is used for both heavy and light

nodes. The number of independent experiment used in the training is 10^5 experiments. The results presented in section VII are based on a grid search that identifies the optimal single and multi-annealing schedules. Building on a single-annealing schedule baseline, PAOA introduces a heterogeneous scheduling strategy by partitioning the graph into two groups: one with large coupling strengths (“heavy” nodes) and another with significantly smaller couplings (“light” nodes). Both groups initially share the same optimized schedule. As shown in FIG. ??, PAOA adaptively modifies this schedule, accelerating the annealing of light nodes while maintaining a slower rate for heavy ones. Despite starting from an optimal schedule, PAOA’s learned adjustment introduces structured heterogeneity, which leads to improved performance, with the average energy per spin moving from $\langle E \rangle / n = -839.2348$ to $\langle E \rangle / n = -839.3907$ success probability of solving 500 instances, showing that the double schedules learned by PAOA have a consistent and a statistical advantage over a single annealing schedule.

Re: NCOMMS-25-58950A
Generalized Probabilistic Approximate Optimization Algorithm

Abdelrahman S. Abdelrahman,¹ Shuvro Chowdhury,¹ Flaviano Morone,² and Kerem Y. Camsari¹

¹*Department of Electrical and Computer Engineering,
University of California, Santa Barbara, Santa Barbara, CA 93106, USA*

²*Center for Quantum Phenomena, Department of Physics,
New York University, New York, New York 10003 USA*

(Dated: November 6, 2025)

=====

I. REVIEWER #2

=====

The authors have substantially revised the manuscript and addressed my comments. I reread the paper and noticed a few minor typos and inconsistencies, that will likely be caught at the proof stage. I did catch one issue. You compute an estimated distribution ρ_f but never define f . Define it on first use (e.g., $f =$ “final layer”?). Admittedly I got side tracked by an interesting observation about figure 5 (d). I spent 3 hours exploring that data. Clearly that was research and not related to the report. I hope many other researchers get side tracked by the wonderful results in this manuscript. I recommend publication without further delay. I am looking forward to reading the published version of the article.

AUTHOR RESPONSE

We thank the reviewer for the comprehensive review that has markedly improved the manuscript.

I did catch one issue. You compute an estimated distribution ρ_f but never define f . Define it on first use (e.g., $f =$ “final layer”?).

AUTHOR RESPONSE/ACTION

We corrected this notational inconsistency by using ρ_p all throughout, where p represents the number of layers and ρ represents the estimated distribution.